# MA-EGOQA: QUESTION ANSWERING OVER EGOCENTRIC VIDEOS FROM MULTIPLE EMBODIED AGENTS

## ABSTRACT

As embodied models become powerful, humans will collaborate with multiple embodied AI agents at their workplace or home in the future. To ensure better communication between human users and the multi-agent system, it is crucial to interpret incoming information from agents in parallel and refer to the appropriate context for each query. Existing challenges are to effectively compress and communicate high volumes of individual sensory inputs in the form of video and to correctly aggregate multiple egocentric videos to construct system-level memory. In this work, we first formally define a novel problem of understanding multiple long-horizon egocentric videos simultaneously collected from embodied agents. To facilitate research in this direction, we introduce MultiAgent-EgoQA (MA-EgoQA), a benchmark designed to systemically evaluate existing models in our scenario. MA-EgoQA provides 1.7k questions unique to multiple egocentric streams, spanning five categories: social interaction, task coordination, theory-of-mind, temporal reasoning, and environmental interaction. We further propose a simple baseline model for MA-EgoQA named EgoMAS, which leverages shared memory across embodied agents and agent-wise dynamic retrieval. Through comprehensive evaluation across diverse baselines and EgoMAS on MA-EgoQA, we find that current approaches are unable to effectively handle multiple egocentric streams, highlighting the need for future advances in this direction.

## 1 INTRODUCTION

Recent advances are increasingly transferring progress from text-based environments to physical embodied domains, such as autonomous driving (Fu et al., 2024; Huang et al., 2024a) and industrial robotics (Zhao et al., 2025; Kannan et al., 2024). Just as large language models like ChatGPT have reshaped how humans interact with and reason over knowledge, we can expect a future where intelligent agents increasingly assist and augment our physical activities. As these agents become common and multiple agents begin to operate within a shared environment, a growing line of research explores how they can interact and collaborate effectively as a system (Feng et al., 2025). Multi-agent system (MAS) can achieve the goals efficiently by decomposing and executing tasks in parallel, and collective reasoning enables agents to leverage broader context and arrive at more optimal solutions (Chen et al., 2024; Hu et al., 2022). At the same time, multi-agent settings pose unique challenges, such as designing robust communication protocols and developing strategies for sub-task generation and allocation. Addressing these challenges is essential for developing embodied agent systems that can operate at scale and under real-world complexities.

However, contextual understanding and question answering (QA) remain largely underexplored in the study of multiple embodied agents. Prior work has predominantly focused on goal-directed algorithms, task allocation, and completion, particularly within the robotics domain (Zhang et al., 2024; Yuan et al., 2025; Guo et al., 2024). While these directions are critical for system operation, effective actions ultimately depend on accurately interpreting event histories, environmental state, and inter-agent communication. This capability is especially critical in QA scenarios, such as when a human manager queries the system to monitor progress. Importantly, since users often do not address a specific agent, the system must be able to integrate experiences across agents and retrieve relevant episodes to provide a correct answer. Such functionality has broad application in the real world, for

example, querying anomalies across multiple bodycam feeds from police officers, or checking when household robots last cleaned the bathroom. As QA is fundamental for making multi-agent systems transparent, controllable, and manageable, it calls for systematic investigation and comprehensive evaluation.

A main challenge in embodied agents video QA lies in managing extremely long egocentric video streams. Embodied agents perceive their surroundings through cameras mounted on the agents themselves, resulting in egocentric perspectives that differ substantially from conventional third-person videos. Although recent video LLMs (Bai et al., 2025; Wang et al., 2025) have been trained on such inputs, the distributional gap between egocentric and general video data remains significant, motivating the continued development of egocentric-specific models (Pei et al., 2024; Tian et al., 2025). Furthermore, because these video streams are captured continuously during operation, models must be capable of reasoning over very long temporal horizons. In realistic scenarios, embodied agents may operate for days, with each agent generating massive video histories, making it difficult to aggregate information and deliver accurate answers. In practice, current video LLMs are still limited to processing videos at the scale of only a few hours (Wang et al., 2024a; Wu et al., 2024), falling short of the requirements for multi-agent QA.

Next, we construct a new benchmark evaluating the models' understanding capability in multi-agent egocentric videos, named MultiAgent-EgoQA (MA-EgoQA). MA-EgoQA consists of 1.7k question-answer pairs based on the EgoLife egocentric video dataset (Yang et al., 2025c). Captured by 6 people living in a shared house for 7 days, the dataset provides five categories unique and fundamental in multi-agent scenarios: social interaction, task coordination, theory of mind, temporal reasoning, and environmental interaction. Questions and answers are generated using GPT-based pipelines, refined through LLM filtering, and ultimately validated by human annotators to ensure quality. We also propose a simple baseline model called EgoMAS, designed to illustrate the potential of multi-agent video QA. EgoMAS aggregates information from all agents into a shared memory, and upon receiving a query, it selectively retrieves the relevant information from the appropriate agents. Compared to the method naively concatenating all information from agents, selective retrieval based on the shared memory of EgoMAS is remarkably token-efficient and superior in QA performance.

We evaluate a range of existing LLMs and video LLMs on MA-EgoQA, as well as our EgoMAS model. Results reveal that current models struggle to handle multiple egocentric video streams effectively: even state-of-the-art LLMs and video LLMs fail to capture the complexities of multi-agent egocentric video understanding. By contrast, EgoMAS, despite its simplicity, achieves performance comparable to Gemini-2.5-Flash (Comanici et al., 2025) operating at a 1M-token context length, using only Qwen2.5-VL-7B (Bai et al., 2025). These findings underscore both the difficulty of the task and the promise of specialized approaches like EgoMAS.

## 2 RELATED WORK

### 2.1 MULTIPLE EMBODIED AGENTS SYSTEM

Existing works on multiple embodied agent systems have been developed, focusing on studying effective approaches to cooperate in diverse environments. CoELA (Zhang et al., 2024) integrates perception, memory, and execution in a modular framework and coordinates plans across agents in a natural language using LLM, and Co-NavGPT (Yu et al., 2023a) employs VLM as a global planner to enable multiple robots to explore complex environments. Meanwhile, several papers evaluate models under limited communication resources and partial observability, which is closer to real-world settings (Yu et al., 2023b; Jain et al., 2020). It is also shown that structured and leadership prompting enhances teamwork efficiency and reduces unnecessary communication (Guo et al., 2024), and PARTNR (Chang et al., 2024) built a large-scale human–robot collaboration benchmark, demonstrating that even state-of-the-art LLM-based systems still face limitations in planning and coordination. However, while these studies have primarily focused on optimizing action execution and cooperation strategies, the problem of integrating egocentric experiences collected over long periods by multiple agents to perform question answering remains insufficiently addressed.

### 2.2 VIDEO QUESTION-ANSWERING BENCHMARKS

A number of video QA benchmarks has been introduced to evaluate the capabilities of video LLMs across diverse dimension. NExT-QA (Xiao et al., 2021) provides 52k video QA samples that focus on

Table 1: Related works for MA-EgoQA dataset.

| | # QA | Duration | Egocentric | Ultra Long | Cross-Video Alignment | Theory-of-Mind |
|---|---|---|---|---|---|---|
| AssistQ (Wong et al., 2022) | 531 | 115 sec | ✓ | ✗ | ✗ | ✗ |
| EgoTextVQA (Zhou et al., 2025) | 7,064 | 102 sec | ✓ | ✗ | ✗ | ✗ |
| EgoMemoria (Ye et al., 2024) | 7,026 | 1-60 min | ✓ | ✗ | ✗ | ✗ |
| MuMA-ToM (Shi et al., 2025) | 900 | 36 sec | ✗ | ✗ | ✗ | ✓ |
| EgoSchema (Mangalam et al., 2023) | 5,063 | 180 sec | ✓ | ✗ | ✗ | ✗ |
| EgoThink (Cheng et al., 2024) | 700 | - | ✓ | ✗ | ✗ | ✓ |
| EgoToM (Li et al., 2025) | 1,039 | 300 sec | ✓ | ✗ | ✗ | ✓ |
| EgoLifeQA (Yang et al., 2025c) | 6,000 | 44 hour | ✓ | ✓ | ✗ | ✗ |
| CVBench (Zhu et al., 2025) | 1,000 | 106 sec | ✗ | ✗ | ✓ | ✗ |
| EgoExoLearn (Huang et al., 2024b) | 2,200 | 13 min | ✓ | ✗ | ✓ | ✗ |
| MA-EgoQA (Ours) | 1,741 | 266 hour | ✓ | ✓ | ✓ | ✓ |

social interaction and temporal reasoning, using 5.4k videos with a duration of 44 seconds in average. Perception Test (Patraucean et al., 2023) offers a holistic evaluation platform which particularly evaluates perception and reasoning ability including understanding of human-object interactions, and uses videos averaging 23 seconds in length. MuMA-ToM (Shi et al., 2025) is designed to test the ability to infer the mental state of individuals in the video. However, these benchmarks are all based on single short videos and do not require cross-video alignment or knowledge fusion across different viewpoints. In contrast, MA-EgoQA uses six videos captured simultaneously from multiple agents, which requires models to build a global understanding of the entire multi-agent system while also tracking each agent's perspective.

### 2.3 Egocentric Video Understanding Benchmarks

Since egocentric video understanding is crucial for real-world applications, a number of benchmarks have been introduced to evaluate video QA in this setting. Ego4D (Grauman et al., 2022) released 3,670 hours of egocentric video along with multiple tasks, including episodic memory and hand–object interaction, establishing a fundamental resource for egocentric video research. Building on this foundation, EgoSchema (Mangalam et al., 2023) evaluates minute-level video understanding with a focus on long-term context, while EgoThink (Cheng et al., 2024) defined six core capabilities and twelve dimensions of egocentric reasoning to assess the interpretive and inferential capacity of VQA models. Extending beyond recognition, EgoPlan-Bench (Chen et al., 2023) introduced tasks for embodied planning, examining how models can connect visual observations with action planning in egocentric settings. EgoTextVQA (Zhou et al., 2025) evaluates a model's ability to understand text information that appears in egocentric videos, while EgoMemoria (Ye et al., 2024) focuses on assessing the recognition and memorization of visual details in egocentric videos. AssistQ (Wong et al., 2022) requires models to watch instructional videos and assist users in completing target actions from an egocentric viewpoint. However, in all of these benchmarks, the video duration per sample remains shorter than one hour, which is not reflective of embodied agents that operate continuously for days.

To address this limitation, EgoLife (Yang et al., 2025c) constructed a super-long egocentric video dataset in which six individuals wore camera-equipped glasses to capture their daily experiences over seven consecutive days in a shared house. While this dataset breaks the length barrier of prior work, its QA benchmark, EgoLifeQA, is designed under a single-agent assumption, and questions can be answered by referencing only one individual's memory. As a result, it still does not evaluate the capability of aggregating and aligning experiences across agents. Meanwhile, MA-EgoQA is the first to evaluate the QA task on multiple, super-long, and temporally aligned egocentric videos.

## 3 MA-EgoQA Benchmark

We now introduce MA-EgoQA, a new benchmark designed to evaluate the ability of models to comprehend and reason over multiple egocentric video streams from embodied agents. We first formally define the task in Sec. 3.1. Next, Secs. 3.3, 3.4, and 3.5 describe the dataset construction pipeline, including sample generation, LLM-based filtering, and human validation. Finally, in Sec. 3.6, we provide a detailed analysis of MA-EgoQA, presenting dataset statistics and qualitative examples.

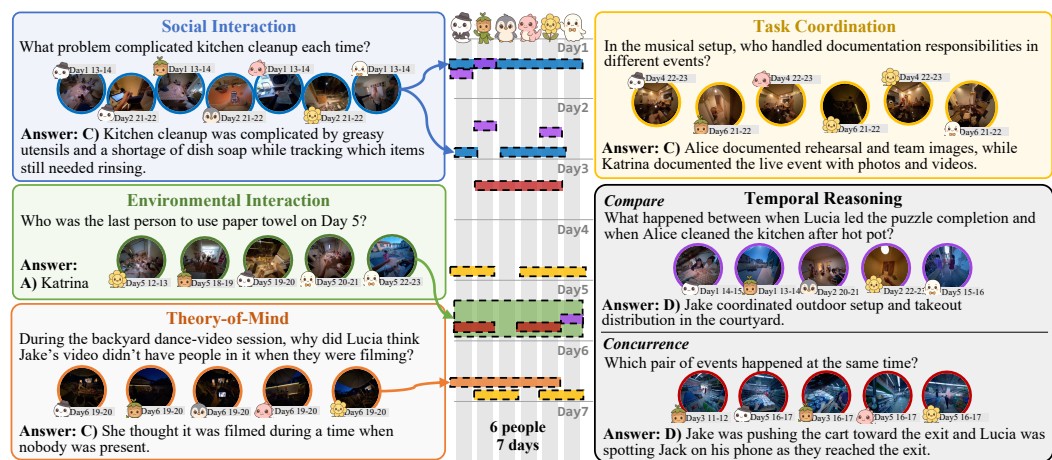

Figure 1: **Examples from MA-EgoQA** across five categories and their subcategories. MA-EgoQA is the first multiple embodied agents egocentric video QA benchmark, requiring comprehension of six egocentric videos spanning seven days per query. All categories are unique to multi-agent settings.

## 3.1 TASK DEFINITION

We formally define the **Multi-Agent Egocentric Video Question Answering** task as follows. Let there be $N$ embodied agents, each continuously recording egocentric video during a time span of $T$ days. The video stream of agent $i$ is denoted by

$$V_i = \{(f_i^1, t_i^1), (f_i^2, t_i^2), \ldots, (f_i^{M_i}, t_i^{M_i})\},$$

where $f_i^j$ represents the $j$-th clip observed by agent $i$ at timestamp $t_i^j$, and $M_i$ is the total number of frames recorded by agent $i$. The complete multi-agent video corpus is then $\mathcal{V} = \{V_1, V_2, \ldots, V_N\}$, which in our benchmark corresponds to $N = 6$ agents over $T = 7$ days of continuous recording. Given a natural language query $q \in \mathcal{Q}$, the system must generate an answer $a \in \mathcal{A}$ such that $a = \mathcal{F}(q, \mathcal{V})$, where $\mathcal{F}$ denotes a reasoning function that jointly processes visual evidence and temporal information from all agents. Importantly, the answer must be grounded in the multi-agent egocentric streams $a^* = \arg\max_{a \in \mathcal{A}} \Pr(a \mid q, \mathcal{V})$, where $a^*$ denotes the ground-truth answer validated by human annotators.

## 3.2 BENCHMARK CATEGORIES

We design five categories in **MA-EgoQA benchmark** prior to generating the question-answer pairs, ensuring that the data generation pipeline is optimized for each. In selecting these categories, our primary goal is to capture aspects that are unique to the multi-agent setting and critical for human-agent system interactions in real-world. The categories are described as follows.

**Social Interaction (SI)**  This category evaluates the ability to accurately localize and ground casual conversations or affiliative behaviors across the video streams. This category contains questions related to how people engage or respond to others, group behaviors without specific goal sequences involving multiple people interacting, and topics or meaningful information during the conversation.

**Task Coordination (TC)**  As highlighted in Sec. 2.1, research on MAS has largely focused on collaboration to achieve shared goals. To reflect this practical importance, we define this category. Questions in this category address how roles were assigned, responsibilities divided, and actions sequenced toward goal completion, and how decisions were made throughout the execution of tasks.

**Theory of Mind (ToM)**  Theory of Mind refers to the cognitive ability to reason about the mental states of others, including their thoughts, beliefs, desires, and emotions. Effective use of diverse perspectives is crucial for capturing contextual cues and improving performance on ToM questions. This category includes queries about what an agent believed or misunderstood, what information they could or could not perceive, and the intentions underlying their actions and speech.

**Temporal Reasoning (TR)**  Accurate understanding of multi-agent experiences requires aligning the timelines of different egocentric video streams into a coherent global view. To evaluate these abilities, the TR category is divided into two subcategories: concurrence, which focuses on what one

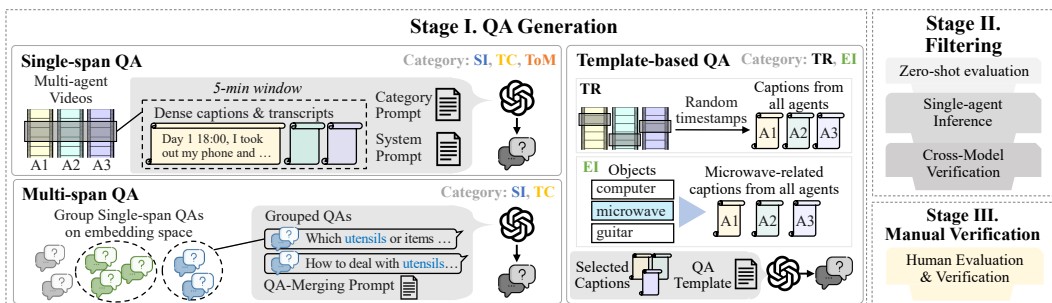

Figure 2: Benchmark construction pipeline. QA pairs are first generated for each category through its dedicated process. These pairs are subsequently refined through LLM filtering and human evaluation.

agent was doing while others were engaged in different activities, and temporal comparison, which examines the relative ordering of events.

**Environmental Interaction (EI)** Agents also engage with their surrounding environment, such as operating a vacuum cleaner or turning on a faucet. Because these interactions are distributed across multiple agents, aggregating them is essential for tracking environmental states and planning appropriate actions to achieve shared goals. Questions in this category explore object usage, including frequency, first-time use, and which agent engaged with them most.

### 3.3 DATA GENERATION

After defining the five categories of MA-EgoQA, we construct candidate question–answer pairs using GPT-based generation. Since the nature of questions differs across categories, we adopt two complementary strategies. For SI, TC, and ToM, where question formats are diverse and open-ended, we generate a large pool of samples and subsequently filter low-quality pairs using the LLM-based pipeline described in Sec. 3.4. In contrast, for TR and EI, which typically involve more structured queries, we design predefined templates and generate question–answer pairs by instantiating them with contextual information. For convenient evaluation, MA-EgoQA is designed to be multiple-choice questions with five options. Below, we describe the candidate generation process in detail.

**Single Span Question Generation** Let the continuous egocentric video streams be partitioned into fixed-length windows of $\Delta = 5$ minutes. For each category $c \in \{\text{SI}, \text{TC}, \text{ToM}\}$, we denote the set of video windows as: $\mathcal{W}_c = \{w_1^c, w_2^c, \ldots, w_{N_c}^c\}$. Each window $w_i^c$ is associated with textual information $\mathcal{T}(w_i^c)$, which consists of captions and transcripts within that interval: $\mathcal{T}(w_i^c) = \{\text{captions}, \text{transcripts}\}$. Given the system prompt $\pi_{\text{sys}}$ and a category-specific prompt $\pi_c$, the input to the data generator is defined as $x_i^c = \big(\mathcal{T}(w_i^c), \pi_{\text{sys}}, \pi_c\big)$. The generator model $G$ (GPT-4o in our implementation) produces a structured output:

$$G(x_i^c) \mapsto \big(q_i^c, a_i^c, \{f_{i,j}^c\}_{j=1}^4, r_i^c, \tau_i^c\big),$$

where $q_i^c$ is the generated question, $a_i^c$ is the correct answer, $\{f_{i,j}^c\}_{j=1}^4$ are four distractor options, $r_i^c$ is the rationale for the correct answer, and $\tau_i^c$ are the referred timestamps.

Finally, the dataset for category $c$ is the union of all generated samples:

$$\mathcal{D}_c = \bigcup_{i=1}^{N_c} \big(q_i^c, a_i^c, \{f_{i,j}^c\}_{j=1}^4, r_i^c, \tau_i^c\big).$$

In total, the constructed datasets contain $|\mathcal{D}_{\text{SI}}| = 33.4\text{k}$, $|\mathcal{D}_{\text{TC}}| = 31.6\text{k}$, and $|\mathcal{D}_{\text{ToM}}| = 34.1\text{k}$ samples.

**Multi-span Question Generation** To extend MA-EgoQA into a truly long-context benchmark, we generate multi-span questions that require reasoning over multiple non-contiguous temporal windows. Formally, let the set of single-span QA pairs be $\mathcal{D} = \{(q_i, a_i)\}_{i=1}^M$, where $q_i$ and $a_i$ denote the question and its corresponding answer, respectively. For each pair, we compute an embedding using a text encoder $E(\cdot)$: $e_i = E\big([q_i; a_i]\big) \in \mathbb{R}^d$.

We then define pairwise similarity between two samples as $s_{ij} = \langle e_i, e_j \rangle / \|e_i\|\|e_j\|$. A similarity graph $G = (V, E)$ is constructed where each vertex corresponds to a QA pair, and an edge $(i, j)$

exists if $s_{ij} \geq \delta$, with $\delta$ a predefined threshold. Connected components $\{\mathcal{C}_1, \ldots, \mathcal{C}_K\}$ are extracted from this graph, where each component contains at least two semantically related QA pairs.

For each component $\mathcal{C}_k$, a multi-span QA pair is synthesized by merging the single-span questions and answers:

$$(q_k^{\text{multi}}, a_k^{\text{multi}}) = G\big(\{(q_i, a_i) \mid i \in \mathcal{C}_k\}\big),$$

where $G(\cdot)$(GPT-5) denotes the generation process that consolidates multiple related events into a single long-context query. At this stage, only the question and its correct answer are generated. To ensure difficulty and quality, false options are created separately, rather than being directly inherited from the single-span samples. The resulting multi-span dataset is defined as:

$$\mathcal{D}_{\text{multi}} = \{(q_k^{\text{multi}}, a_k^{\text{multi}}, \text{options}_k)\}_{k=1}^K,$$

where each multi-span question requires referencing multiple temporal spans to answer correctly. We generated 15.9k samples for SI multi-span and 16.3k samples for TC multi-span questions.

**Template based Question Generation**   For the TR category, we formalize the data generation process as follows. Let the set of captions for agent $a \in \mathcal{A}$ be denoted by $\mathcal{C}_a = \{(t, c_{a,t}) \mid t \in \mathcal{T}_a\}$, where $t$ is a timestamp and $c_{a,t}$ is the caption associated with $a$ at time $t$. Captions are pre-processed at three levels of temporal granularity $\Delta \in \{30\text{s}, 10\text{min}, 1\text{h}\}$.

For the time comparison subcategory, we randomly sample five timestamps $\{t_1, \ldots, t_5\}$ from an agent $a$. To avoid redundancy, we enforce unique agent sampling whenever possible. The selected captions are $\{c_{a,t_j} \mid j = 1, \ldots, 5\}$. Given a predefined prompt template $p \in \mathcal{P}_{\text{comp}}, |\mathcal{P}_{\text{comp}}| = 4$, the generator model $G$ produces a QA sample:

$$G(p, \{c_{a,t_j}\}_{j=1}^5) \mapsto (q, \{o_1, \ldots, o_5\}, y),$$

where $q$ is the natural-language question, $\{o_i\}_{i=1}^5$ are candidate options, and $y \in \{1, \ldots, 5\}$ is the index of the correct answer. For the concurrence subcategory, we sample a timestamp $t$ and collect captions from multiple agents $\mathcal{A}_t \subseteq \mathcal{A}$: $\{c_{a,t} \mid a \in \mathcal{A}_t\}$. Conditioned on a prompt template $p \in \mathcal{P}_{\text{conc}}, |\mathcal{P}_{\text{conc}}| = 3$, the generator outputs:

$$G(p, \{c_{a,t}\}_{a \in \mathcal{A}_t}) \mapsto (q, \{o_1, \ldots, o_5\}, y),$$

where $q$ is formulated to ask about event concurrence across agents. Final generated TR dataset consists of two subcategories: $\mathcal{D}_{\text{TR}} = \mathcal{D}_{\text{comp}} \cup \mathcal{D}_{\text{conc}}$, where each element is a triple $(q, \{o_1, \ldots, o_5\}, y)$.

For the EI category, we first enumerate a vocabulary of environment objects and agent-initiated actions. Let $\mathcal{V} = \{v_1, v_2, \ldots, v_{|\mathcal{V}|}\}$ denote the vocabulary consisting of both environment objects and agent-initiated actions. For each day $d \in \mathcal{D}$, we collect dense captions from all agents, denoted as $\mathcal{C}_d = \{c_{u,t} \mid u \in \mathcal{U}, t \in T_d\}$, where $u$ indexes agents and $t$ timestamps. Given a target item $v \in \mathcal{V}$, we aggregate all captions from day $d$ that mention $v$: $\mathcal{C}_d(v) = \{c_{u,t} \in \mathcal{C}_d \mid v \in c_{u,t}\}$. Conditioned on this aggregated context and an EI-specific instruction prompt $\pi_{\text{EI}}$, the generative model $G$ (GPT-5) produces a question–answer pair $G\big(\mathcal{C}_d(v), \pi_{\text{EI}}\big) \mapsto (q_d^v, a_d^v)$. The final EI dataset is thus defined as

$$\mathcal{D}_{\text{EI}} = \bigcup_{d \in \mathcal{D}} \bigcup_{v \in \mathcal{V}} (q_d^v, a_d^v).$$

## 3.4   LLM FILTERING

To ensure that only high-quality and challenging samples progress to the human validation stage, we employ an automatic filtering procedure based on large language models. Specifically, each generated question–answer pair is verified by prompting an LLM with a restricted input context and evaluating whether the model can correctly infer the answer. If the model can easily predict the correct answer under these constraints, the sample is deemed insufficiently challenging and excluded from the benchmark. All filtering steps are conducted using GPT-5 in the following order, which serves as the backbone for quality control in our data generation pipeline.

**Zero-shot Filtering**   In this stage, each question is evaluated without providing any contextual information, thereby testing whether the correct answer can be trivially inferred. If the model consistently predicts the correct option under such a zero-context setting, the sample is considered unchallenging and excluded. Concretely, we query the model three times per question, and if it produces the correct answer in more than two trials, the corresponding sample is discarded from the dataset.

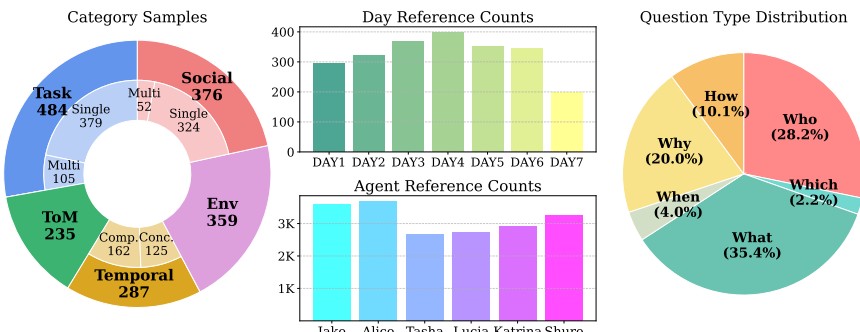

Figure 3: Statistics for MA-EgoQA; *(Left)* number of samples per category; *(Center)* day/agent reference counts of all categories; *(Right)* question type distribution of SI, TC, ToM.

**Single Agent Filtering** MA-EgoQA is intended to specifically evaluate reasoning within a multi-agent context. Therefore, we remove any sample that can be answered correctly using only the memory of a single individual. To enforce this criterion, we extract all human names appearing in the question and the correct answer, and then perform inference using each corresponding individual's memory. If no human name is present, we randomly select one from six available agents and conduct inference with their memory. Whenever the model is able to produce the correct answer under any single-agent memory condition, the sample is eliminated from the benchmark.

**Cross-model Validation** All preceding filtering stages rely on GPT-5, which may introduce model-specific biases. To address this, we include an additional verification step using two external models, Gemini-2.5-flash and Claude-Sonnet-4. Each sample is re-examined under the same contextual information available at generation time. This step not only checks the correctness of the answer but also validates the question and the false options, ensuring that the query is non-trivial. If either external model flags a sample as invalid, it is removed from the dataset.

### 3.5 HUMAN VERIFICATION

Following all automatic filtering steps, a final human verification stage is conducted to ensure the quality and validity of the remaining candidates. Human verifiers are granted access to the complete set of information, including captions, transcripts, and videos from all agents, as well as the validation rationales produced by Gemini and Claude. Each verifier evaluates samples according to the same guidelines used during data generation, ensuring consistency with the original design criteria. In total, four human verifiers reviewed 3,436 candidates, from which 1,741 high-quality samples were selected to form the MA-EgoQA benchmark.

### 3.6 BENCHMARK ANALYSIS

We analyze the distribution of questions in MA-EgoQA between different types, categories, days, and agents, and present the statistics in Figure 3. We observe that the questions are well distributed across the five categories, the seven recorded days, and the six agents. This balance ensures that the evaluation is not dominated by a single skill or event but provides a diverse set of challenges and requires the model to track and reason about all agents effectively. The questions in the benchmark also exhibit natural linguistic diversity. The majority of questions begin with "What" and "Who", while "Why", "How", "When", and "Which" questions provide additional variation. Overall, the broad coverage of questions in MA-EgoQA ensures a comprehensive evaluation of multimodal reasoning in egocentric multi-agent scenarios.

## 4 EGOCENTRIC VIDEO UNDERSTANDING IN MULTI-AGENT SYSTEM

Beyond introducing the new benchmark, we also propose a simple yet effective training-free baseline, named EgoMAS (Egocentric Multi-Agent System), to stimulate future research on MA-EgoQA. EgoMAS is a centralized MAS that is specifically designed to tackle the unique challenges of multi-agent egocentric reasoning. First, it leverages an event-based shared memory that enables a system-level global understanding by integrating fragmented events across agents. Second, EgoMAS dynamically selects which agent's memory to reference and adapts the search query for each agent, allowing fine-grained reasoning across multiple perspectives. A more detailed description of EgoMAS is provided in the following section.

Table 2: Evaluation results of 16 baselines and EgoMAS on MA-EgoQA.

| Generator | Context len | Social Interaction | | | Task Coordination | | | Theory of Mind | Temporal Reasoning | | | Environmental Interaction | Avg |
|---|---|---|---|---|---|---|---|---|---|---|---|---|---|
| | | Single | Multi | All | Single | Multi | All | | Compare | Concur. | All | | |
| Random | | 20.00 | 20.00 | 20.00 | 20.00 | 20.00 | 20.00 | 20.00 | 20.00 | 20.00 | 20.00 | 20.00 | 20.00 |
| Gemini-2.5-flash | 1M | 40.43 | 46.15 | 41.22 | 38.79 | 27.62 | 36.36 | 24.26 | 43.21 | 51.28 | 46.59 | 33.98 | **36.93** |
| Llama-3.1-Nemotron-8B | 1M | 20.68 | 19.23 | 20.48 | 22.43 | 24.84 | 22.93 | 20.43 | 20.37 | 25.60 | 22.65 | 21.17 | 21.65 |
| Qwen2.5-7B-Instruct-1M | 500k | 27.78 | 23.08 | 27.13 | 28.23 | 31.43 | 28.93 | 20.85 | 20.99 | 28.00 | 24.04 | 26.18 | 26.08 |
| GPT-5 | 272k | 37.65 | 26.92 | 36.17 | 35.62 | 27.62 | 33.88 | 22.55 | 37.65 | 42.40 | 39.71 | 38.72 | 34.81 |
| Qwen3-30b-a3b-instruct | 240k | 30.25 | 25.00 | 29.52 | 27.18 | 29.52 | 27.69 | 16.60 | 18.52 | 28.80 | 23.00 | 26.46 | 25.56 |
| gpt-oss-120b | 128k | 30.86 | 15.38 | 28.72 | 29.55 | 20.00 | 27.48 | 22.55 | 30.25 | 25.60 | 28.22 | 25.63 | 26.82 |
| gpt-oss-20b | 128k | 19.44 | 17.31 | 19.15 | 20.08 | 20.95 | 28.10 | 23.83 | 26.54 | 27.70 | 26.83 | 25.35 | 24.81 |
| VideoChat-Flash (10k f) | 32k | 24.07 | 21.15 | 23.67 | 26.65 | 19.05 | 25.00 | 20.00 | 22.22 | 27.20 | 24.39 | 20.73 | 23.06 |
| VideoXL-2 (4k f) | 32k | 20.06 | 21.15 | 20.21 | 17.41 | 28.57 | 19.83 | 19.15 | 20.59 | 20.80 | 20.91 | 21.73 | 20.39 |
| Qwen2.5-VL-7B (1.9k f) | 128k | 26.54 | 25.00 | 26.33 | 25.07 | 31.43 | 26.45 | 21.28 | 22.22 | 28.80 | 25.09 | 25.07 | 25.22 |
| InternVideo2 | - | 24.38 | 26.92 | 24.73 | 27.97 | 18.10 | 25.83 | 18.30 | 21.60 | 24.75 | 22.81 | 27.86 | 24.52 |
| Qwen2.5-VL-7B (EgoRAG) | - | 22.53 | 21.15 | 22.34 | 21.64 | 19.05 | 21.07 | 18.30 | 18.52 | 11.20 | 15.33 | 18.87 | 19.57 |
| Ego-R1 | - | 23.46 | 15.38 | 22.34 | 27.97 | 17.14 | 25.62 | 18.30 | 22.22 | 32.80 | 26.83 | 28.41 | 24.70 |
| Qwen2.5-VL-7B (BM25) | - | 41.05 | 23.08 | 38.56 | 39.84 | 18.10 | 35.12 | 31.91 | 22.84 | 38.40 | 29.62 | 30.08 | 33.49 |
| Qwen2.5-VL-7B (DPR) | - | 26.85 | 26.92 | 26.86 | 28.23 | 22.86 | 27.07 | 21.28 | 21.60 | 40.00 | 29.62 | 20.33 | 25.27 |
| EgoMAS (Text) | - | 40.74 | 19.23 | 37.77 | 43.01 | 22.86 | 38.64 | 25.96 | 30.86 | 38.40 | 34.15 | 36.49 | 35.55 |
| EgoMAS (Text + Video) | - | 42.90 | 15.38 | 39.10 | 41.69 | 24.76 | 38.02 | 24.68 | 30.86 | 40.80 | 35.19 | 37.88 | _35.96_ |

## 4.1 EVENT-BASED SHARED MEMORY

At every 10-minute interval, each embodied agent provides a caption summarizing its own observations during the preceding time span. A centralized manager then integrates these individual captions into a system-level summary. Rather than producing a flat textual condensation, the manager first identifies key events across agents and, for each event, explicitly records the corresponding 4W1H fields: When, What, Where, Who, and How, producing a coherent global memory that aligns agent perspectives while preserving critical details for multi-agent reasoning.

## 4.2 AGENT-WISE DYNAMIC RETRIEVAL

Given a query $q$, EgoMAS first retrieves top-$n$ system-level memories from the shared memory $\mathcal{M}_{\text{shared}}$ using BM25 ranking $\mathcal{R}_{\text{sys}}(q) = \text{Top-}n \ \{(m, s(m, q)) \mid m \in \mathcal{M}_{\text{shared}}\}$, where $s(m, q)$ denotes the BM25 similarity score between memory $m$ and query $q$. From the retrieved system-level context, EgoMAS generates a set of agent-specific retrieval requests $\mathcal{Q}_{\text{agent}} = \{(a_j, q_j)\}_{j=1}^J$, where each request consists of an agent identifier $a_j$ and a sub-query $q_j$. For each $(a_j, q_j)$, EgoMAS performs agent-level retrieval from the agent's memory $\mathcal{M}_{a_j}$:

$$\mathcal{R}_{a_j}(q_j) = \text{Top-}k \ \{(m, s(m, q_j)) \mid m \in \mathcal{M}_{a_j}\}.$$

To ensure relevance, we filter out memories with scores below a predefined threshold $\tau$:

$$\widetilde{\mathcal{R}}_{a_j}(q_j) = \{(m, s(m, q_j)) \in \mathcal{R}_{a_j}(q_j) \mid s(m, q_j) \geq \tau\}.$$

Finally, the centralized manager conditions on the aggregated filtered results $\widetilde{\mathcal{R}} = \bigcup_{j=1}^J \widetilde{\mathcal{R}}_{a_j}(q_j)$ to generate the final natural-language response $\hat{y} = F(q, \widetilde{\mathcal{R}})$, where $F$ denotes the generation function.

## 5 EXPERIMENTAL SETUPS

We evaluate a range of competitive open-source and proprietary LLMs and video LLMs on MA-EgoQA. Proprietary LLMs include Gemini-2.5-flash (Comanici et al., 2025) and GPT-5 (OpenAI, 2025). Open-source LLMs include Llama-3.1 (Grattafiori et al., 2024), Qwen2.5-1M (Yang et al., 2025b), Qwen3 (Yang et al., 2025a), and gpt-oss (Agarwal et al., 2025). Video LLMs include VideoChat-Flash (Li et al., 2024), Video-XL-2 (Qin et al., 2025), and Qwen2.5-VL (Bai et al., 2025).

For text-only LLMs, we concatenate captions from all agents in chronological order and directly pass them to the LLM along with the question. Long-context models (Gemini-2.5-Flash, Llama-3.1-Nemotron-8B, Qwen2.5-7B-Instruct-1M) use 10-min captions, while general text-only models (GPT-5, Qwen3-30b-a3b-Instruct, gpt-oss-120b, gpt-oss-20b) use 1-hr captions. For video LLMs, we concatenate all 30-second clips from all agents in chronological order and uniformly sample the number of frames that each model can receive without any error. VideoChat-Flash leverages 10k frames, while VideoXL-2 and Qwen2.5-VL-7B take 4k and 1.9k frames, respectively.

In addition to these simple baselines, we also evaluate several retrieval-based baselines, including simple text-based retrieval methods BM25 (Robertson et al., 2009) and DPR (Karpukhin et al., 2020)

Table 3: Ablation study on EgoMAS structure.

| Shared Memory | Agent-wise Dyn. Ret. | Acc. |
|:---:|:---:|:---:|
| ✗ | ✗ | 27.80 |
| ✗ | ✓ | 28.20 |
| ✓ | ✗ | 30.04 |
| ✓ | ✓ | **35.55** |

Table 4: Ablation studies on sub-modules of EgoMAS.

(a) Shared Memory Structure

| Shared Memory Structure | Acc. |
|:---|:---:|
| Summary | 30.67 |
| Triplet | 30.44 |
| Chunk | 25.96 |
| Graph | 31.99 |
| Ours (4W1H) | **35.55** |

(b) Memory Retriever

| Memory Retriever | Acc. |
|:---|:---:|
| DPR | 28.67 |
| Qwen3-Embed-0.6B | 33.03 |
| NV-Embed-v2 7B | **37.91** |
| Ours (BM25) | 35.55 |

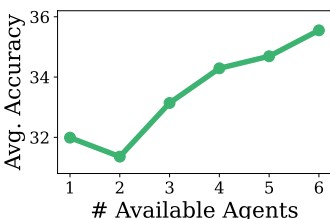

Figure 4: Accuracy of Ego-MAS with limited agents.

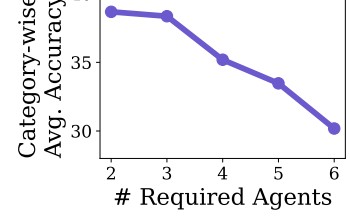

Figure 5: EgoMAS accuracy with the number of gold agents.

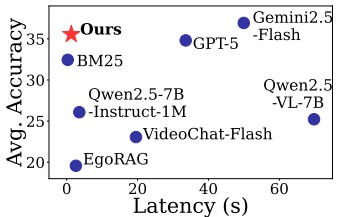

Figure 6: Comparison of inference latency and accuracy.

with Qwen2.5-VL and text feature-based retrieval with InternVideo2 (Wang et al., 2024b). Lastly, there are two baselines that work on the EgoLife video dataset, but in a single-agent setting. Since they do not support multi-agent scenarios, we only use videos from a single person in EgoRAG (Yang et al., 2025c) (and generate answers using the GPT-4o model), and randomly select one person when Ego-R1 (Tian et al., 2025) performs reasoning.

For EgoMAS, we evaluate two variants: EgoMAS (Text) and EgoMAS (Text+Video). Both models retrieve relevant clips based on the textual information of videos, but in the response generation stage, EgoMAS (Text) only uses text captions, but EgoMAS (Text+Video) uses captions and 8 video frames.

# 6 EXPERIMENTAL RESULTS AND ANALYSIS

Table 2 summarizes the performance of a diverse set of models, as well as our proposed EgoMAS method, on MA-EgoQA. In general, we observe that MA-EgoQA is a highly challenging benchmark. Even the strongest model, Gemini-2.5-flash, achieves only an average accuracy of 36.93%. In particular, several well-known open source models, such as Llama-3.1 and VideoXL-2, perform only marginally above random chance, underscoring the difficulty of MA-EgoQA. Among the five evaluation categories, Theory of Mind emerges as the most challenging, with all models struggling to move beyond the low-20s in accuracy. Moreover, in both Social Interaction and Task Coordination, the multi-agent setting is substantially harder than the single-agent setting, revealing the added complexity of reasoning over the memory of multiple agents. The difficulty of theory of mind and multi-agent reasoning represent the core novelties of MA-EgoQA compared to prior benchmarks.

**Performance of EgoMAS** Compared to other methods based on Qwen2.5-VL-7B, EgoMAS demonstrates consistent improvements across different categories of MA-EgoQA. EgoMAS with only text outperforms traditional text retrieval methods BM25 and DPR. EgoMAS with both text and video outperforms the frame dump baseline by 11.74% on average, significantly narrowing the gap to Gemini-2.5-flash to less than 1% while using a much smaller model. The superior performance of EgoMAS highlights the strength of selective retrieval on a shared memory across all agents.

Figure 6 shows the average latency measured over 100 randomly selected MA-EgoQA samples and overall accuracy of baseline models and EgoMAS (Text). Non-retrieval-based models show substantially higher computational overhead, whereas retrieval-based models, including EgoMAS, exhibit notably lower latency. In particular, EgoMAS achieves the highest accuracy among retrieval-based models while requiring only 1.3 seconds per query. These results highlight that EgoMAS is a practical approach for building a multi-agent egocentric video QA system in real world settings.

**Efficacy of Two Methods in EgoMAS** To validate the efficacy of each method in EgoMAS, we perform an ablation study using our (Text) model, as shown in Table 3. For the model without shared

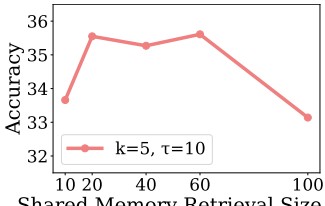 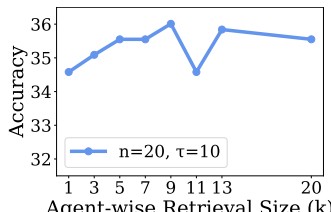 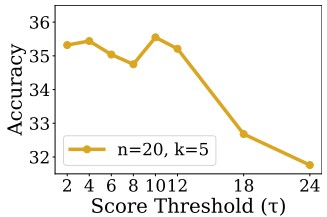

Figure 7: Ablation studies on shared memory retrieval size ($n$), agent-wise dynamic retrieval size ($k$), and score threshold ($\tau$).

memory, we use 10-minute captions for all agents as a retrieval source. Meanwhile, the models without agent-wise dynamic retrieval leverage 10 minute retrieval results from the previous step and generate the response. Using both methods together achieves the highest accuracy, highlighting aggregating memories from all agents into shared memory is effective and dynamic retrieval method helps the model to find the most informative and related knowledge from the corpus.

**Analysis on sub-modules of EgoMAS**    We validate the contribution of each sub-module with ablation studies in Table 4. In Table 4a, we evaluate multiple memory construction strategies introduced in prior works (Zeng et al., 2024; Gutiérrez et al., 2025) and 4W1H memory. The event-based 4W1H memory structure clearly outperforms alternative methods, demonstrating its strong ability to abstract and fuse the events across multiple agents. Table 4b presents a comparison between our choice of memory retriever and other dense retrievers (Zhang et al., 2025; Lee et al., 2024). Although NV-Embed-v2 achieves the highest accuracy, it contains 7B parameters and consequently incurs substantial computational and temporal overhead. In contrast, BM25 employs lightweight keyword-based retrieval and delivers competitive performance, highlighting its practicality.

**Analysis on Hyperparameters in EgoMAS**    There are three hyperparameters in EgoMAS: shared memory retrieval size ($n$), agent-wise dynamic retrieval size ($k$), and score threshold ($\tau$). In Figure 7, we report the performance of EgoMAS with different values of the hyperparameter. Shared memory retrieval size is optimal around 20 to 60, since if it is too small or big, then models suffer from finding meaningful information from the given context. However, the model shows robust performance over $k$, while high threshold $\tau$ decreases the accuracy by filtering most of the retrieved data.

**Analysis on sensitivity to the number of available agents**    We investigate the sensitivity of MA-EgoQA to the number of agents by limiting the available agent views based on EgoMAS performance. As shown in Figure 4, the results show a clear trend that using more agents leads to better accuracy from 31.99% to 35.55%. These results highlights that MA-EgoQA indeed requires models to incorporate information from multiple agents. In Figure 5, we illustrate the performance breakdown of EgoMAS based on the number of agents whose memory is required to answer the question. We observe a clear downward trend in the average accuracy as more agents are involved. Specifically, the average accuracy is 38.68% when only two agents are related to the answer and 30.18% when six agents are involved in the answer. These results indicate that retrieving relevant information from multiple agents and constructing a coherent global understanding from individual views is the primary source of difficulty in MA-EgoQA.

## 7 CONCLUSION

In this work, we present MA-EgoQA, a novel benchmark for question answering over multiple long-horizon egocentric video streams from embodied agents, spanning five categories that capture the core challenges of multi-agent reasoning. Alongside, we propose EgoMAS, a simple training-free baseline that combines shared memory with agent-wise dynamic retrieval, achieving competitive performance to recent frontier models with significantly long context despite using a much smaller model. Our evaluations show that current LLMs and video LLMs struggle with multi-agent egocentric setting, particularly in the theory of mind and multi-agent reasoning, underscoring the complexity of the task and highlighting MA-EgoQA as a promising future direction for embodied AI research.

## ETHICS STATEMENT

Since MA-EgoQA discusses multi-agent settings over egocentric video, if the model is used in the real-world, users should be cautious on privacy problems. Moreover, sharing personal information with others in multi-agent setting takes responsibility. We recommend the users to apply this framework on embodied robots first and check any possible harms.

## REPRODUCIBILITY STATEMENT

We check the reproducibility of evaluation scores on MA-EgoQA multiple times. All details on data generation and evaluation setup are described in Section 3 and Section 3.3. The video dataset (EgoLife) is currently open-sourced and MA-EgoQA will also be open-sourced with the evaluation code.

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

## A  LLM USAGE

We use LLM in data generation and polishing sentences during paper writing (such as grammar checking). Final benchmarks are verified by human verifiers and we state that LLM did not play a significant role.

## B  LIMITATION AND FUTURE WORK

This work has limitations as follows. First, while EgoLife provides 266 hours of egocentric videos from six people that cover a wide range of daily activities and environments, MA-EgoQA is based only on EgoLife and does not include other scenarios in different environments. This is because EgoLife is currently the only pubilcly available video dataset providing long-term, egocentric videos captured simultaneously from multiple agents. We hope more video datasets proposed in this direction and evaluate the model in various types of videos to ensure the generalization. Next, regarding the small performance gap between EgoMAS (Text) and EgoMAS (Text+Video), there is room to improve the utilization of visual information of EgoMAS. One promising research direction is adaptive modality selection during inference. For example, the model could rely on video frames when a query depends on fine-grained visual details and use textual captions when questions can be answered from abstracted information alone. We are excited to see more future works on how to leverage visual information from multiple agents effectively.

## C  BENCHMARK QA SAMPLES

We present example questions and options for each category from the next page, with the correct answers shown in bold.

Table 5: Social Interaction (SI) - Multi Span category samples

| Category | Social Interaction, Multi Span |
| --- | --- |
| Question | **How did egg tart impressions shift between first sight and first bite?**
(A) Alice mistook the egg tart for a muffin, then Shure found the egg tart tasty despite looks.
**(B) Jake mistook the egg tart for a pancake, then Shure found the egg tart tasty despite looks.**
(C) Alice mistook the egg tart for a muffin, then Shure found the egg tart tasty despite looks.
(D) Jake mistook the egg tart for a pancake, then Tasha found the egg tart bland despite looks.
(E) Alice mistook the egg tart for a muffin, then Tasha found the egg tart bland despite looks.

Source: Jake (Day5 8PM), Alice (Day5 8PM), Lucia (Day5 8PM), Shure (Day6 5PM) |
| Contextual Videos | 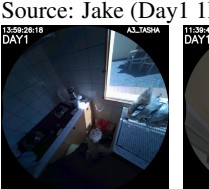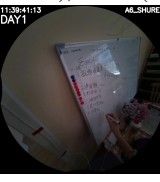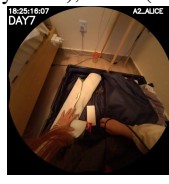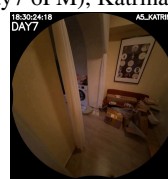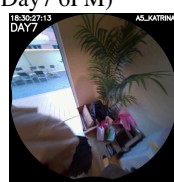 |
| Question | **What coffee discussions happened in both situations?**
(A) Jake proposed restocking coffee beans and equipment and later Lucia offered packing help after Lucia mentioned coffee.
**(B) Shure proposed restocking coffee beans and equipment, and later Alice offered packing help after Katrina mentioned coffee.**
(C) Jake proposed restocking coffee beans and equipment, and later Lucia offered packing help after Lucia mentioned coffee.
(D) Jake proposed restocking coffee beans and equipment, and later Alice offered packing help after Katrina mentioned coffee.
F(E) Shure proposed restocking coffee beans and equipment, and later Lucia offered packing help after Lucia mentioned coffee.

Source: Jake (Day1 1PM), Shure (Day 1PM), Alice (Day7 6PM), Katrina (Day7 6PM) |
| Contextual Videos | |

Table 6: Social Interaction (SI) - Single Span category samples

| Category | Social Interaction, Single Span |
|---|---|
| **Question** | **Who helped each other find scissors, and what were the scissors used for?**
(A) Jake lent scissors to Shure, who used them for prepping coffee beans.
**(B) Jake handed scissors to Katrina for her package and flower crafts.**
(C) Tasha and Lucia found scissors to cut plastic wrappings for food.
(D) Alice shared scissors with Katrina for cutting papers.
(E) Lucia requested scissors from Tasha for assembling wooden shelves

Source: Jake (Day4 12PM), Katrina (Day4 12 PM) |
| **Contextual Videos** |  |
| **Question** | **Who asked for a power bank and how did Jake respond to their request?**
(A) Lucia borrowed a power bank, but Jake didnŽ2019t have any left.
(B) Nicous asked for a power bank, but Jake refused to let him unplug anything.
(C) Shure asked for a power bank, and Jake offered heavy assistance.
**(D) Tasha borrowed a power bank, and Jake offered to find another one.**
(E) Alice needed a power bank, and Jake handed his own immediately.

Source: Jake (Day6 12PM) |
| **Contextual Videos** |  |

Table 7: Task Coordination (TC) - Multi span category samples

| Category | **Task Coordination, Multi Span** |
|---|---|
| **Question** | **In the flower tasks, what roles did participants take in different events?**
(A) Jake observed and asked clarifying questions, later arranged the bouquets, while Katrina handled hands-on trimming and shaping.
(B) Alice observed and asked clarifying questions, later asked about flower types, while Katrina handled hands-on trimming and shaping.
**(C) Jake observed and asked clarifying questions, later asked about flower types, while Shure handled hands-on trimming and shaping.**
(D) Alice observed and asked clarifying questions, later asked about flower types, while Shure handled hands-on trimming and shaping.
(E) Jake observed and asked clarifying questions, later asked about flower types, while Katrina handled hands-on trimming and shaping.

Source: Jake (Day1 11AM), JAKE (Day2 4PM), Shure (Day4 4PM) |
| **Contextual Videos** | 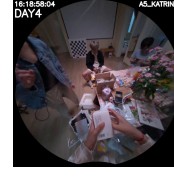 |
| **Question** | **In the delivery task, who coordinated unpacking and organizing, and who later managed packing in a different event?**
(A) Alice coordinated unpacking and organizing, and later Lucia managed packing the delivery.
**(B) Lucia and Katrina coordinated unpacking and organizing, and later on Jake managed packing the delivery.**
(C) Alice coordinated unpacking and organizing, and later Jake managed packing the delivery.
(D) Lucia and Katrina coordinated unpacking and organizing, and later Jake managed packing the delivery.
(E) Lucia and Katrina coordinated unpacking and organizing, and later Lucia managed packing the delivery."

Source: Lucia (Day4 4PM), Katrina (Day4 4PM) |
| **Contextual Videos** | 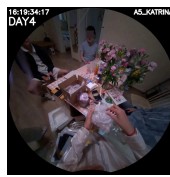 |

Table 8: Task Coordination (TC) - Single span category samples

| Category | **Task Coordination, Single Span** |
| --- | --- |
| **Question** | **How did they resolve concerns about the safety of the power strip?**
(A) Lucia conducted a test and decided it was fine for usage.
(B) Shure insisted safety wasnŽ2019t a big issue and moved forward without discussion.
(C) Shure reassured everyone it was brand-new and safe to use.
(D) Katrina brought over a different power strip for better safety.
**(E) Jake and Alice discussed using a reputable brand and reassured others.** |

Source: Jake (Day2 6PM), Alice (Day 6PM)

| | |
| --- | --- |
| **Contextual Videos** | 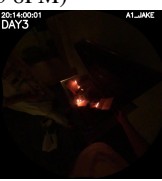 |

| **Question** | **When they were trying to light the charcoal, how did the group coordinate and divide tasks to make it work?**
**(A) Jake shielded the fire with a cardboard while Shure carefully transported it outside.**
(B) Alice ignited the fire using dry leaves while Katrina monitored the process.
(C) Lucia set up a fan to make the fire stronger while Tasha prepared water for safety.
(D) Jake and Lucia used chemicals to ignite the charcoal while Alice held a bucket nearby.
(E) Tasha managed the cooking station while Shure brought wood for the fire. |
| --- | --- |

Source: Jake (Day3 8PM)

| | |
| --- | --- |
| **Contextual Videos** | 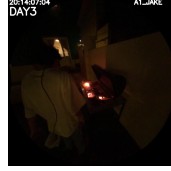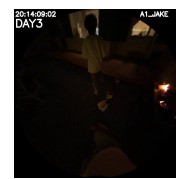 |

Table 9: Theory of Mind (ToM) category samples

| Category | Theory of Mind |
|---|---|
| **Question** | **What did Katrina wrongly assume while looking at Nicous and Violet filming?**
**(A) That they were wrapping up their session immediately.**
(B) That Jake wasn't involved in their filming plans.
(C) That they were photographing desserts.
(D) That filming had been done in the second-floor living room.
(E) That Violet and Nicous would soon ask for her help.

Source: Jake (Day6 10PM), Katrina (Day6 10PM) |
| **Contextual Videos** |  |
| **Question** | **Why did Shure think Choiszt might have misinterpreted the coffee instructions?**
(A) Choiszt thought they needed a latte machine instead of an espresso machine.
(B) Choiszt added too much water and ignored exact measurements.
**(C) Choiszt confused the steps and thought milk was required.**
(D) Choiszt started brewing with a wrong type of coffee grinder.
(E) Choiszt assumed fruity coffee beans required special brewing techniques.

Source: Shure (Day2 3PM) |
| **Contextual Videos** |  |
| **Question** | **Why didn't Katrina know where to put the big kitchen tools?**
(A) Tasha was using the drawer for other purposes.
(B) Katrina wasn't involved in cleaning the kitchen before.
**(C) Shure had placed things randomly, making it hard to locate spaces.**
(D) The storage arrangement in the kitchen had recently changed.
(E) Alice moved things while reorganizing the kitchen.

Source: Katrina (Day4 11AM) |
| **Contextual Videos** |  |

Table 10: Temporal Reasoning (TR) - Compare category samples

| Category | **Temporal Reasoning, Compare** |
|---|---|
| Question | **What happened between when Lucia led the puzzle completion and supply planning and when Alice deep-cleaned the kitchen after hot pot?**
(A) Shure repaired a drone at dawn before anyone arrived.
(B) Tasha oversaw final errands and device data consolidation discussions.
(C) Lucia ran a midnight karaoke session after the meeting.
**(D) Jake coordinated outdoor setup and takeout distribution in the courtyard.**
(E) Katrina led themed room planning and decor decisions.

Source: Jake (Day1 2PM), Alice (Day2 8PM), Lucia (Day1 1PM) |
| Contextual Videos |  |
| Question | **Which is the correct sequence of events?**
A) Alice retrieved a power bank, doubled back for a forgotten item, and then took a ride-hailing car.
B) Tasha wrapped up checking tickets and led a group discussion on May Day travel, even demonstrating a bus ticket booking.
C) Jake set up lighting and equipment for meal prep and then worked on computer file transfers.
(A) A-C-B
(B) B-C-A
(C) A-B-C
**(D) C-A-B**
(E) B-A-C

Source: Jake (Day2 6PM), Alice (Day4 9PM), Tasha (Day2 11AM) |
| Contextual Videos |  |

Table 11: Temporal Reasoning (TR) - Concurrence category samples

| Category | Temporal Reasoning, Concurrence |
| --- | --- |
| Question | **Which pair of events happened at around the same time?**
(A) Alice was checking out at the cashier and Shure was paying for bottled water at the self-checkout.
(B) Lucia was loading boxes into a car and Jake was waiting outside the store.
(C) Tasha was standing in line at the bakery counter and Katrina was asking a clerk about discounts.
**(D) Jake was pushing the cart toward the exit and Lucia was spotting Jack on his phone as they reached the exit.**
(E) Shure was sitting on a bench making a phone call and Alice was driving to the store.

Source: Jake (Day5 4PM), Tasha (Day5 4PM) |
| Contextual Videos |  |
| Question | **When Katrina was fertilizing the flowers, and Alice was checking the dilution and water amounts, what was Jake doing?**
(A) Pouring the nutrient solution into the vases himself
(B) Going upstairs to do his nails and charge his phone.
(C) Assembling a display rack by the whiteboard and taking inventory.
**(D) Wiping down plates and the table with new wet wipes while tidying.**
(E) Stepping outside to take all the trash out to the bin.

Source: Jake (Day1 7PM), Alice (Day1 7PM), Katrina (Day1 7PM) |
| Contextual Videos |  |

Table 12: Environmental Interaction (EI) category samples

| Category | Environmental Interaction |
| --- | --- |
| Question | **Who used the microwave the most on DAY3?**
(A) Katrina
**(B) Tasha**
(C) Lucia
(D) Jake
(E) Shure |
| Question | **When was the first time guitar was used on DAY2?**
(A) 6 PM
(B) 5 PM
(C) 4 PM
(D) 9 PM
**(E) 3 PM** |
| Question | **How many people used oven on DAY1?**
**(A) 4**
(B) 5
(C) 1
(D) 2
(E) 3 |

