# OpenReview forum: "MA-EgoQA: Question Answering over Egocentric Videos  from Multiple Embodied Agents"
_ICLR.cc/2026/Conference — Submitted to ICLR 2026_

### Official Review · Reviewer_oPM5 · 2025-10-31

**Soundness:** 2
**Presentation:** 2
**Contribution:** 2
**Rating:** 2
**Confidence:** 4

**Summary:**

This paper introduces MA-EgoQA, a benchmark dataset for question answering over long-duration, multi-agent egocentric videos. The benchmark is built on the EgoLife dataset, which features 266 hours of video from 6 agents interacting in a shared house over 7 days. The authors' core contribution is a set of 1.7k question-answer pairs specifically designed to be answerable only by aggregating information from multiple agents' video streams. The benchmark spans five challenging, multi-agent categories: Social Interaction, Task Coordination, Theory-of-Mind (ToM), Temporal Reasoning, and Environmental Interaction.

The paper uses a "single-agent filtering" step to remove any questions solvable by one agent's perspective alone. The authors also propose EgoMAS, a training-free baseline model that uses an event-based shared memory and agent-wise dynamic retrieval.

Experimental results, testing over 16 baselines (including SOTA LLMs and Video LLMs), show that MA-EgoQA is challenging. The top model, Gemini-2.5-flash, achieves only 36.93% accuracy (vs. 20% random chance), and all models struggle with the Theory-of-Mind category.

**Strengths:**

- The paper addresses a new and interesting topic of multi-agent egocentric videos, where video streams are captured continuously during operation.

- The data is verified and selected by human annotators.

**Weaknesses:**

- Not enough qualitative analysis. Can the author show and analyze more benchmark samples? One sample is not enough to help audience understand the scope and quality of the benchmark data.

- The performance gain from adding video frames in the EgoMAS (Text+Video) variant over the EgoMAS (Text) variant is very modest (35.96% vs. 35.55%). This suggests either that the text is sufficient for most questions or that the model's method of incorporating video (sampling 8 frames) is not sophisticated enough to be impactful.

**Questions:**

- More qualitative analysis

- How do the VLM models handle long videos in the evaluation of Table 2?

- What is the distribution of video lengths in the benchmark? How does this influence the performance?

---

> ### Author Response · Authors · 2025-11-22
> **Response to Reviewer oPM5 (1/3)**
>
> We deeply appreciate your meaningful and valuable comments. We also greatly thank you for your recognition of our contributions, including 1) the novel problem setting which is multi-agent, long-term egocentric video QA and 2) human verification efforts to control the quality of the benchmark. We have carefully considered your review and made every effort to thoroughly address all your comments and concerns.
>
> ---
> > **Weakness 1)** Not enough qualitative analysis. Can the author show and analyze more benchmark samples? One sample is not enough to help audience understand the scope and quality of the benchmark data.
>
> > **Question 1)** More qualitative analysis
>
> We agree with your comment and appreciate you pointing this out. We present and analyze more benchmark samples for every category in MA-EgoQA below. We omit false options here to provide more examples concisely. We also included these examples in `Appendix C` (Benchmark QA Samples) of the revision to help readers understand our benchmark.
>
> ## Social Interaction
> ```
> (Multi-span)
> Q. How did egg tart impressions shift between first sight and first bite?
> A. Jake mistook the egg tart for a pancake, then Shure found the egg tart tasty despite its looks.
>
> Q. What coffee discussions happened in both situations?
> A. Shure proposed restocking coffee beans and equipment, and later Alice offered packing help after Katrina mentioned coffee.
>
> (Single-span)
> Q. Who helped each other find scissors, and what were the scissors used for?
> A. Jake handed scissors to Katrina for her package and flower crafts.
>
> Q. Who asked for a power bank and how did Jake respond to their request?
> A. Tasha borrowed a power bank, and Jake offered to find another one.
> ```
>
> We include two multi-span and two single-span QA examples in the SI category above. For multi-span examples, the model should refer to multiple agents' memories at different timestamps to answer them correctly, **making the evaluation truly require the knowledge fusion from multiple agents.** Single span examples also demand perception from multiple views at the same time, which is also verified with single-agent filtering.
>
> We would also like to highlight that **false options make the multi-span questions more challenging.** As shown in the example below, we intentionally construct false options to be highly similar to the correct answer, so that **the model cannot respond correctly by relying on only a single memory.** For instance, even if the model retrieves the memory in which Jake mistook the egg tart for a pancake, it still must incorporate a second memory to distinguish between options B and D.
>
> ```
> (Option example)
> A) Alice mistook the egg tart for a muffin, then Shure found the egg tart tasty despite its looks.
> B) Jake mistook the egg tart for a pancake, then Shure found the egg tart tasty despite its looks.
> C) Alice mistook the egg tart for a muffin, then Shure found the egg tart tasty despite its looks.
> D) Jake mistook the egg tart for a pancake, then Tasha found the egg tart bland despite its looks.
> E) Alice mistook the egg tart for a muffin, then Tasha found the egg tart bland despite its looks.
> ```
>
> ## Task Coordination
>
> ```
> (Multi-span)
> Q. In the flower tasks, what roles did participants take in different events?
> A. Jake observed and asked clarifying questions, later asked about flower types, while Shure handled hands-on trimming and shaping.
>
> Q. In the delivery task, who coordinated unpacking and organizing, and who later managed packing in a different event?
> A. Lucia and Katrina coordinated unpacking and organizing, and later Lucia managed packing the delivery.
>
> (Single-span)
> Q. How did they resolve concerns about the safety of the power strip?
> A. Jake and Alice discussed using a reputable brand and reassured others.
>
> Q. When they were trying to light the charcoal, how did the group coordinate and divide tasks to make it work?
> A. Jake shielded the fire with a cardboard while Shure carefully transported it outside.
> ```
>
> We present four examples for the TC category, covering both multi-span and single-span settings. In the multi-span examples, we qualitatively verify that the questions require long-horizon, task-related memories from multiple agents. Importantly, **the relevant actions appear only in each individual agent's memory, thus constructing a global memory by integrating all agents' viewpoints is essential.** These questions focus on task allocation in multi-agent collaboration, which is a key aspect of real-world multi-agent systems and highlights the practicality of the MA-EgoQA benchmark.
>
> Beyond task allocation, the single-span examples demonstrate that **the questions can also target grounded, task-related discussions.** For instance, the first single-span example requires extracting meaningful information from conversations rather than relying solely on action observations.

---

> ### Author Response · Authors · 2025-11-22
> **Response to Reviewer oPM5 (2/3)**
>
> ## Theory-of-Mind
> ```
> Q. What did Katrina wrongly assume while looking at Nicous and Violet filming?
> A. That they were wrapping up their session immediately.
>
> Q. Why did Shure think Choiszt might have misinterpreted the coffee instructions?
> A. Choiszt confused the steps and thought milk was required.
>
> Q. Why didn't Katrina know where to put the big kitchen tools?
> A. Shure had placed things randomly, making it hard to locate spaces.
> ```
>
> We provide three examples of ToM questions above. In the first example, the model must contrast Katrina's memory with the actual filming scenario involving Nicous and Violet, identify the inconsistency between their perspectives, and infer the resulting mistaken belief. **This demonstrates that ToM questions require deeper reasoning based on the integration of multi-agent memories.** In the second example, Choiszt’s misunderstanding of the coffee instructions is never explicitly stated, and the model needs to infer it from subtle cues in the interaction with Shure. In the final example, Katrina’s uncertainty about where to place the kitchen tools is only implicitly conveyed and must be linked to Shure’s earlier behavior of placing items arbitrarily. **Collectively, these cases highlight that ToM questions demand an additional layer of inference that goes beyond what is directly visible or spoken.**
>
> ## Temporal Reasoning
> ```
> (Compare)
> Q. What happened between when Lucia led the puzzle completion and supply planning and when Alice deep-cleaned the kitchen after the hot pot?
> A. Jake coordinated outdoor setup and takeout distribution in the courtyard.
>
> Q. Which is the correct sequence of events?
> A) Alice retrieved a power bank, doubled back for a forgotten item, and then took a ride-hailing car.
> B) Tasha wrapped up checking tickets and led a group discussion on May Day travel, even demonstrating a bus ticket booking.
> C) Jake set up lighting and equipment for meal prep and then worked on computer file transfers.
> A. C-A-B
>
> (Concurrence)
> Q. Which pair of events happened at around the same time?
> A. Jake was pushing the cart toward the exit, and Lucia was spotting Jack on his phone as they reached the exit.
>
> Q. When Katrina was fertilizing the flowers, and Alice was checking the dilution and water amounts, what was Jake doing?
> A. Wiping down plates and the table with new wet wipes while tidying
> ```
>
> These are examples of Compare and Concurrence questions in the TR category. In examples in Compare, the model should locate each event from multiple agents and compare the timestamps to get the correct timeline among them. **These questions directly evaluate the temporal alignment capability of the models, which is fundamental in constructing a multi-agent system.** Moreover, this alignment becomes more difficult since MA-EgoQA is based on extremely long-horizon videos spanning 7 days. In Concurrence examples, questions further emphasize temporal alignment in multi-agent systems, focusing on the events at the same timestamp. Please notice that **each event can happen in separate places, showing that the questions cannot be answered with a single agent memory.**
>
> ## Environmental Interaction
> ```
> Q. Who used the microwave the most on Day 3?
> A. Tasha
>
> Q. When was the first time a guitar was used on Day 2?
> A. 3 PM
>
> Q. How many people used an oven on Day 1?
> A. 4
> ```
> Lastly, we present the example questions from the EI category. These examples clearly illustrate that EI questions focus on specific objects in the environment within a given day, which requires a capability distinct from the other categories. In particular, **the model must track interactions between agents and their surroundings and construct system-level observation** by integrating the views of multiple agents. If the model does not incorporate each agent’s memory, it is likely to produce incorrect answers for many questions, such as counting how many people used a particular object or identifying the first time an object was used.

---

> ### Author Response · Authors · 2025-11-22
> **Response to Reviewer oPM5 (3/3)**
>
> > **Weakness 2)** The performance gain from adding video frames in the EgoMAS (Text+Video) variant over the EgoMAS (Text) variant is very modest (35.96% vs. 35.55%). This suggests either that the text is sufficient for most questions or that the model's method of incorporating video (sampling 8 frames) is not sophisticated enough to be impactful.
>
> We fully understand your concern and thank you for raising this point. Regarding the small performance gap between EgoMAS (Text) and EgoMAS (Text+Video), we believe this is because **the generated captions already include partial descriptions of visual events, such as actions and visual cues.** As a result, the additional video frames sometimes provide overlapping information.
>
> However, given that EgoMAS (Text+Video) achieves only 35.96% while surpassing most baselines, we fully agree that there is room for improving the way visual information is utilized. We believe that more sophisticated visual integration methods could unlock further performance gains. In fact, as shown in `Table 2`, the performance gains of EgoMAS (Text+Video) over EgoMAS (Text) in SI (37.77% to 39.10%), TR (34.15% to 35.19%), and EI (36.49% to 37.88%) indicate that **visual signals do contribute meaningful improvement.**
>
> Motivated by these findings, one promising research direction is adaptive modality selection during inference. For example, the model could rely on video frames when a query depends on fine-grained visual details and use textual captions when questions can be answered from abstracted information alone.
>
> As the first work to introduce multi-agent egocentric video QA and the MA-EgoQA benchmark, **our goal is to propose EgoMAS as a simple, training-free, yet effective baseline that already outperforms most existing approaches.** We are excited about future work that can build on this foundation, and we hope EgoMAS will serve as a useful starting point for advancing multi-agent video understanding.
>
> We include this discussion in `Appendix B` (Limitation and Future Work) of the revision.
>
> ---
> > **Question 2)** How do the VLM models handle long videos in the evaluation of Table 2?
>
> Thanks for raising the question on the evaluation settings of baselines and giving us the opportunity to clarify the experimental details. For uniform sampling baselines (VideoChat-Flash, VideoXL-2, Qwen2.5-VL-7B), we concatenate all videos from multiple agents into a single long video and uniformly sample the frames as many as each model can process within its input limits, as described in `Section 5` (Experimental Setups). For retrieval-based baselines (InternVideo2, EgoRAG, BM25, DPR), we first retrieve the relevant memories using each retriever, and feed the frames and their corresponding captions, along with the query, as input to Qwen2.5-VL-7B.
>
> ---
> > **Question 3)** What is the distribution of video lengths in the benchmark? How does this influence the performance?
>
> We would like to clarify that every question in MA-EgoQA is evaluated using the full video set (266 hours in total), thus there is no difference in video length across questions. Each agent has approximately 44 hours of egocentric video, and the benchmark includes six agents. We adopt this evaluation setup because multi-agent egocentric video QA aims to assess a model’s ability to interpret multiple long-duration videos captured by different agents who operate concurrently within the same environment.

---

> > ### Comment · Reviewer_oPM5 · 2025-11-27
> >
> > Thank you for the feedback, which has solved parts of my concerns. I have raised my rating to 4.
> >
> > Could you explain more about how typical VLM models, such as Qwen-VL (even Gemini), are effectively able to process a 266-hour video in a single pass? Even with thousands of frames, are they able to capture the visual details to answer questions like "Who asked for a power bank?" If they cannot, is it more reasonable to cut the long videos into shorter ones as a more reasonable evaluation setting?
> >
> > For the qualitative analysis, I also recommend including the relevant visual frames alongside the questions—especially those directly related to them. Without these, it’s difficult to assess whether the questions and answers are reasonable.

---

> > > ### Author Response · Authors · 2025-12-03
> > > **Response to Reviewer oPM5**
> > >
> > > We sincerely appreciate your thoughtful feedback and are glad to provide further clarification on our evaluation settings. Our evaluation includes three categories of baselines: 1) models using all captions from all agents, 2) models using all frames from all agents, and 3) retrieval-based models.
> > >
> > > 1. **All Captions models** (second group in `Table 2`, Gemini-2.5-flash to gpt-oss-20b):
> > > These models receive concatenated captions from all agents along with the query, without retrieval. Specifically, we provide concatenated 10-minute captions to Gemini-2.5-flash, Llama-3.1-Nemotron-8B, and Qwen2.5-7B-Instruct-1M, and concatenated 1-hour captions to GPT-5, Qwen3-30b-a3b-instruct, and gpt-oss models, according to their respective context length limits.
> > > 2. **All Frames models** (third group in `Table 2`, VideoChat-Flash to Qwen2.5-VL-7B (1.9k f)):
> > > For these baselines, we concatenate all frames from all agents and input them into the model with the query. Frames are uniformly sampled up to each model’s processing limit.
> > > 3. **Retrieval-based models** (fourth group in `Table 2`, InternVideo2 to Qwen2.5-VL-7B (DPR)):
> > > These models divide long videos into smaller segments and retrieve the most relevant ones for each query. This setting aligns more closely with practical and scalable long-video understanding.
> > >
> > > As you correctly observed, non-retrieval-based models (the first two categories) struggle with extremely long inputs since it is difficult for them to capture the information required for accurate reasoning. Our findings reinforce that a retrieval-based approach is essential for effectively handling such lengthy, multi-agent video data. Thus, we include retrieval-based baselines to offer a fair and realistic comparison with EgoMAS, which also retrieves relevant short clips rather than processing the entire video at once.
> > >
> > > Regarding your suggestion for qualitative analysis, we included the key visual frames alongside the question samples in `Appendix C`. We thank your question and comment again.

---

### Official Review · Reviewer_8MQJ · 2025-10-31

**Soundness:** 2
**Presentation:** 2
**Contribution:** 2
**Rating:** 4
**Confidence:** 5

**Summary:**

The paper introduces a novel MA-EgoQA benchmark designed for multi-agent egocentric video question answering. It aims to evaluate collaborative understanding and reasoning among multiple embodied agents. Each question in the dataset is designed to depend on multiple agents’visual observations, temporal relations, and mental states. The authors further propose a training-free multi-agent framework with shared memory and dynamic retrieval to perform system-level reasoning over multi-agent video data.

**Strengths:**

1. MA-EgoQA is the first benchmark to target multi-agent collaborative reasoning in egocentric settings, which extends beyond single-agent VQA and VideoQA tasks.
2. The dataset integrates multiple synchronized first-person videos and corresponding QA pairs that require cross-agent temporal and social reasoning.
3. It provides a unified evaluation platform for multi-agent perception, memory sharing, and cross-view reasoning, a meaningful direction for embodied AI research.
4. The proposed EgoMAS baseline introduces a reasonable training-free design based on shared memory and agent-wise retrieval, showing competitive performance with larger commercial models.

**Weaknesses:**

1. Current experiments mainly compare different models’overall performance but do not disentangle the sources of task difficulty (e.g., long-horizon temporal reasoning, multi-agent information fusion, or multi-modal dependency).
2. Although the paper claims that ToM questions rely on dialogue and semantic context, it provides no quantitative evidence showing the impact of transcribed speech versus visual-only inputs.
3. There is no sensitivity analysis on the number of agents used. It remains unclear how performance changes when reasoning over fewer or more viewpoints, leaving the true impact of multi-agent data unverified.
4. The shared-memory and dynamic-retrieval modules are not compared with simpler baselines (e.g., uniform sampling, or event-based summarization), which weakens the argument for their necessity.
5. The design of ToM questions likely depends heavily on ASR transcripts, given the absence of raw audio or gaze data. The extent of this dependency is not analyzed, raising uncertainty about whether these tasks truly reflect visuo-cognitive reasoning or are predominantly text-driven.
6. The following datasets are missing from the background or comparisons:

[R1] Mm-ego: Towards building egocentric multimodal LLMs, ICLR, 2025.

[R2] Egotextvqa: Towards egocentric scene-text aware video question answering, CVPR, 2025.

[R3] Assistq: Affordance-centric question-driven task completion for egocentric assistant, ECCV, 2022.

**Questions:**

Please refer to the weaknesses.

---

> ### Author Response · Authors · 2025-11-22
> **Response to Reviewer 8MQJ (1/3)**
>
> We greatly appreciate your insightful and detailed comments. We also thank you very much for your kind recognition of our contributions, including 1) introducing MA-EgoQA as the first benchmark for multi-agent egocentric video reasoning, 2) providing a holistic evaluation platform for meaningful factors of embodied AI research such as multi-agent perception and memory sharing, and 3) demonstrating the efficiency and effectiveness of EgoMAS which is a train-free, simple yet effective baseline for MA-EgoQA. We have carefully considered your review and made every effort to faithfully address all your comments and concerns.
>
> ---
> > **Weakness 1)** Current experiments mainly compare different models’overall performance but do not disentangle the sources of task difficulty (e.g., long-horizon temporal reasoning, multi-agent information fusion, or multi-modal dependency).
>
> Thank you for raising this point. To better understand the sources of difficulty in MA-EgoQA, we break down the performance of EgoMAS along the three aspects you suggested. Our analysis shows that the primary challenge arises from multi-agent information fusion. The detailed results and analysis are presented below.
>
> **[Multi-agent information fusion]**
> In `Table C.1`, we analyze the performance of EgoMAS based on the number of agents whose memory is required to answer the question. We observe **a clear downward trend in the average accuracy as more agents are involved.** In particular, the EI category shows a dramatic drop from 41.51% when two agents are needed to 23.21% when six agents must be referenced. These results indicate that **retrieving relevant information from multiple agents and constructing a coherent global understanding from individual views is the primary source of difficulty in MA-EgoQA.**
>
> **Table C.1.** Analysis of EgoMAS performance with respect to the number of required agents.
> |Category \ # Agents|2|3|4|5|6|
> |-|:-:|:-:|:-:|:-:|:-:|
> |SI|47.22|33.33 | 38.64 | 43.10 | 32.43 |
> |TC|34.15|44.00 | 31.82 | 39.84 | 40.09 |
> |ToM|31.82|28.57 | 33.33 | 22.35 | 25.00 |
> |EI|41.51|47.54 | 36.96 | 28.57 | 23.21 |
> |**Avg**| **38.68** | **38.36** | **35.19** | **33.47** | **30.18** |
>
> **[Multi-modal dependency]** Next, we investigate the modality dependency of EgoMAS in `Table C.2`. For every category, EgoMAS (Text) outperforms EgoMAS (Video), suggesting that textual abstraction provides more effective information than raw frames during inference. **When comparing EgoMAS (Text+Video) with EgoMAS (Text), we observe small but consistent improvements in several categories**, including Social Interaction, Temporal Reasoning, and Environmental Interaction. These gains are reasonable because Temporal Reasoning and Environmental Interaction questions often focus on specific actions or objects that are better captured visually. Although the current improvements from visual information are modest, we believe there is **significant potential for future work to strengthen visual utilization in multi-agent egocentric systems.**
>
>
> **Table C.2.** Analysis of EgoMAS performance across different input modalities.
> | | SI | TC | ToM | TR | EI | Avg |
> |:-:|:-:|:-:|:-|:-|:-:|:-:|
> | EgoMAS (Text) | 37.77  | **38.64**  | **25.96** | 34.15  | 36.49  | 35.55   |
> | EgoMAS (Video) | 31.38  | 35.33  | 23.83 | 32.06  | 36.21  | 32.57   |
> | EgoMAS (Text+Video) | **39.10** | 38.02  | 24.68 | **35.19** | **37.88** | **35.96** |
> | $\Delta_{(T+V) - (T)}$ | 1.33 | -0.62 | -1.28 | 1.04 | 1.39 | 0.41 |
>
> **[Other sources]** Additionally, we analyze the performance of EgoMAS based on the temporal range of the ground-truth timestamps in MA-EgoQA. While questions with the longest temporal span show lower accuracy than those with shorter spans, the variation across ranges is relatively large. This is likely not due to the benchmark itself, but rather because EgoMAS is a retrieval-based method that does not provide the entire long video into the model. In fact, when we evaluate Gemini-2.5-Flash, the average accuracy on SI and TC questions decreases as the number of timestamps increases (from 39.55% to 31.58% when moving from one timestamp to more than three).
>
> We included the accuracy results with respect to the number of required agents in `Figure 5` and its discussion in `Section 6` of the revision to demonstrate the source of task difficulty in MA-EgoQA.

---

> ### Author Response · Authors · 2025-11-22
> **Response to Reviewer 8MQJ (2/3)**
>
> > **Weakness 2)** Although the paper claims that ToM questions rely on dialogue and semantic context, it provides no quantitative evidence showing the impact of transcribed speech versus visual-only inputs.
>
> We appreciate your constructive comment. To validate whether ToM questions rely on dialogue and semantic context, we evaluate the performance of EgoMAS on the ToM category under different input modalities. As shown in `Table C.4`, **the highest accuracy is achieved when captions and transcripts are provided, indicating that ToM questions depend on conversational and semantic cues.**
>
> **Table C.4.** Accuracy of ToM questions across different input modalities.
> | Models | ToM Acc   |
> |-|-|
> | Text (Caption + Transcript) | **25.96** |
> | Visual only | 23.83 |
> | Both | 24.68 |
>
> We also include a qualitative example that further demonstrates this dependency. In the following case, the answer can only be inferred by understanding Alice’s repeated questions about the purpose of the event, showing that **dialogue information is essential for answering many ToM questions.**
>
> ```
> Q. Who was not aware that the tasting event included mineral water brands?
> A. Alice asked multiple times about the purpose of the event before understanding it was a tasting.
> ```
> ---
> > **Weakness 3)** There is no sensitivity analysis on the number of agents used. It remains unclear how performance changes when reasoning over fewer or more viewpoints, leaving the true impact of multi-agent data unverified.
>
> Thank you for raising this important point, which was not sufficiently addressed in our work. We investigate the sensitivity to the number of agents by limiting the available agent views. As shown in `Table C.5`, there is a clear trend that **using more agents leads to better accuracy, highlighting that MA-EgoQA indeed requires models to incorporate information from multiple agents.**
>
> We included these results and analysis on the sensitivity of the number of agents in `Figure 4` and `Section 6` of the revision.
>
> **Table C.5.** Performance of EgoMAS with different numbers of available agents.
> | # Available Agent | Acc   |
> |-|-|
> | 1 | 31.99 |
> | 2 | 31.36 |
> | 3 | 33.14 |
> | 4 | 34.29 |
> | 5 | 34.69 |
> | 6 | **35.55** |
>
> ---
> > **Weakness 4)** The shared-memory and dynamic-retrieval modules are not compared with simpler baselines (e.g., uniform sampling, or event-based summarization), which weakens the argument for their necessity
>
> We understand your concern and thank you for raising this. We would like to kindly remind you that `Table 3` (Ablation study on EgoMAS structure) shows the effectiveness of each module with ablation studies. To demonstrate their necessity more clearly, we provide further analysis using simpler baselines for each module as follows.
>
> In `Table C.6`, we compare different memory construction strategies introduced in prior works [1, 2]. **The event-based shared memory achieves the highest accuracy among all tested baselines,** indicating its strong ability to fuse information from multiple agents.
>
> **Table C.6.** Performance of EgoMAS with different shared memory structures.
> | Shared Memory Structure | Acc   |
> |-|-|
> | Summary | 30.67 |
> | Triplet | 30.44 |
> | Chunk | 25.96 |
> | Graph | 31.99 |
> | Ours (Event-based 4W1H Shared Memory) | **35.55** |
>
> To further emphasize the necessity of agent-wise dynamic retrieval, `Table C.7` reorganizes existing results from `Table 2` in the paper. The "uniform sampling without retrieval" baseline corresponds to the "Qwen2.5-VL-7B (1.9k f)" setting, which uniformly samples frames from all agents. The "agent-agnostic retrieval with BM25" corresponds to "Qwen2.5-VL-7B (BM25)", which retrieves from a unified corpus containing memories from all agents. **Compared to non-retrieval and non-dynamic baselines, our retrieval method shows the best accuracy, highlighting its advantage in MA-EgoQA.**
>
> **Table C.7.** Performance of EgoMAS with different retrieval strategies.
> | Retrieval Method | Retrieval | Dynamic$^*$ | Acc   |
> |-|-|-|-|
> | Uniform sampling w/o retrieval | X | X | 25.22 |
> | Agent-agnostic retrieval w/ BM25 | O | X | 33.49 |
> | Ours (Agent-wise Dynamic Retrieval) | O | O | **35.55** |
>
> $^*$ Dynamic denotes agent-wise dynamic query generation.
>
> [1] Zeng et al., On the structural memory of llm agents, preprint 2024
>
> [2] Gutiérrez et al., From rag to memory: Non-parametric continual learning for large language models, ICML 2025

---

> ### Author Response · Authors · 2025-11-22
> **Response to Reviewer 8MQJ (3/3)**
>
> > **Weakness 5)** The design of ToM questions likely depends heavily on ASR transcripts, given the absence of raw audio or gaze data. The extent of this dependency is not analyzed, raising uncertainty about whether these tasks truly reflect visuo-cognitive reasoning or are predominantly text-driven.
>
> Thank you for your thoughtful comment. We would like to clarify that **QA generation model leverages dense captions together with the transcripts.** **The dense captions provide rich visual information**, including detailed actions (_I am holding my phone facing the dining table._), visual cues (_I watched everyone awkwardly smile._), and even gaze descriptions (_I stared at the front of the house._). **Therefore, the generated ToM questions are grounded not only in dialogue but also in visual details reflected in the captions.**
>
> To further examine whether ToM questions require visual reasoning, we present three examples below. In each case, visual understanding is needed to infer actions (such as Alice touching the grill or Shure placing items) or to recognize spatial relations (such as Katrina being upstairs). **These examples show that, although ToM questions are generated using text, the dense captions encode sufficient visual information to support visuo-cognitive reasoning.**
>
> ```
> Q. Why did Alice assess that the grill was ready while Shure still doubted it?
> A. Alice touched the flame and found it warm enough.
>
> Q. Why didn't Katrina know where to put the big kitchen tools?
> A. Shure had placed things randomly, making it hard to locate spaces.
>
> Q. Why didn’t Katrina notice the glasses charging issue earlier?
> A. She was upstairs in another room at the time.
> ```
>
> ---
> > **Weakness 6)** The following datasets are missing from the background or comparisons: Mm-ego, Egotextvqa, Assistq.
>
> We greatly appreciate your suggestion on related works. MA-EgoQA is the first benchmark that aims to evaluate 1) multi-agent, 2) extremely long, and 3) egocentric video QA. While the mentioned datasets target egocentric video understanding, they are neither multi-agent nor based on days-long videos, which highlights the novelty and necessity of MA-EgoQA. We included comparisons between the mentioned benchmarks and MA-EgoQA in `Table 1` and `Section 2.3` (Egocentric Video Understanding Benchmarks) of the revision.

---

### Official Review · Reviewer_AiKH · 2025-11-03

**Soundness:** 4
**Presentation:** 4
**Contribution:** 3
**Rating:** 8
**Confidence:** 4

**Summary:**

This paper introduces **MA-EgoQA**, a multi-agent egocentric VideoQA benchmark designed to advance research on multi-agent collaboration and human–robot communication. The dataset is built upon **EgoLife**, featuring six actors (agents) and seven days of continuous life-log recordings. The question–answer pairs are automatically generated to cover six aspects: social interaction, task coordination, theory of mind, temporal reasoning, and environmental interaction. To address this task, the authors propose the **EgoMAS** framework, which incorporates a shared memory module and a system-to-individual retrieval mechanism. Experimental results demonstrate that EgoMAS consistently outperforms all baseline methods.

**Strengths:**

1.	MA-EgoQA: The first multi-agent egocentric VideoQA benchmark covering multiple multi-agent relevant QA tasks.
2.	EgoMAS: a simple but reasonable solution framework that achieves superior performance than strong baselines.
3.	Comprehensive baseline analyses, with well-structured presentation.

**Weaknesses:**

1.	The biggest concern in my side is that the questions seem to favor text-based information for answering, as indicated in Table 2. For example, EgoMAS, which utilizes both text and video inputs, only achieves a 0.4% improvement over its text-only counterpart. I also observe that the QA pairs are generated from captions and transcripts. Such text sources inherently lack fine-grained visual details and 3D spatial cues that are critical for embodied agents to understand and navigate indoor environments. I recommend the authors discuss this limitation in the paper.

2.	Except for its egocentric aspect, both multi-agent theory-of-mind [1], social interaction and temporal reasoning [2], environmental interaction [3] are featured in previous datasets. There is a lack of related discussion and comparison in the paper—what are the differences (new challenges) of such questions in MA-EgoQA and in previous datasets?

[1] Shi H, Ye S, Fang X, et al. Muma-tom: Multi-modal multi-agent theory of mind[C]//Proceedings of the AAAI Conference on Artificial Intelligence. 2025, 39(2): 1510-1519.

[2] Xiao J, Shang X, Yao A, et al. Next-qa: Next phase of question-answering to explaining temporal actions[C]//Proceedings of the IEEE/CVF conference on computer vision and pattern recognition. 2021: 9777-9786.

[3] Patraucean V, Smaira L, Gupta A, et al. Perception test: A diagnostic benchmark for multimodal video models[J]. Advances in Neural Information Processing Systems, 2023, 36: 42748-42761.

**Questions:**

Will the dataset be released?

---

> ### Author Response · Authors · 2025-11-22
> **Response to Reviewer AiKH**
>
> We thank you very much for your constructive and meaningful comments. We also sincerely appreciate your recognition of our contributions that 1) MA-EgoQA is the first multi-agent egocentric video QA benchmark, 2) EgoMAS is a simple but highly effective method that outperforms many strong baselines, and 3) comprehensive evaluation with clear presentation. We made every effort to address all your concerns.
>
> ---
> > **Weakness 1)** Questions seem to favor text-based information for answering. EgoMAS, which utilizes both text and video inputs, only achieves a 0.4% improvement over its text-only counterpart. QA pairs are generated from captions and transcripts, which inherently lack fine-grained visual details and 3D spatial cues that are critical for embodied agents to understand and navigate indoor environments. I recommend the authors discuss this limitation in the paper.
>
> Thank you for your thoughtful feedback. We would like to clarify that **many questions in MA-EgoQA rely on information that appears only in the video frames**, especially in the Temporal Reasoning and Environmental Interaction categories. For instance,
> ```
> Q. Which pair of events happened at around the same time?
> A. Tasha was grabbing some tissues from the table and Alice was switching to a thinner brush to start her eye makeup
> ```
> In this question, both actions (grabbing tissues and switching the brush) are identifiable only through the video, not from transcripts, showing that **visual information is essential for answering certain questions.**
>
> Regarding the small performance gap between EgoMAS (Text) and EgoMAS (Text+Video), we believe this is because **the generated captions already contain some descriptions of many visual events, including actions and visual cues.** As a result, visual information might overlap with textual information. Nevertheless, it is clear that there is room to improve the utilization of visual information of EgoMAS, and we are excited to see more future works on how to leverage visual information from multiple agents effectively.
>
> We fully agree that fine-grained visual details and 3D spatial cues are important for embodied agents. This benchmark focuses on multi-agent knowledge fusion and user-system question answering, and richer visual grounding and spatial reasoning are important next steps.
>
> As you mentioned, we included this discussion in `Appendix B` (Limitation and Future Work) of the revision.
>
> ---
> > **Weakness 2)** Except for its egocentric aspect, both multi-agent theory-of-mind, social interaction and temporal reasoning, environmental interaction are featured in previous datasets. There is a lack of related discussion and comparison in the paper—what are the differences (new challenges) of such questions in MA-EgoQA and in previous datasets?
>
> We appreciate you for pointing out the insufficient discussion on the differences between MA-EgoQA and previous datasets. We would like to clarify that, beyond the egocentric aspect, the key differences lie in two points: **1) the use of multiple videos from multi-agent and 2) the extremely long duration of the video streams.** MuMA-ToM asks ToM questions on a single short video that is 36 seconds long on average. NExT-QA contains social interaction and temporal reasoning questions, but each query is still based on a single short video of 44 seconds on average. The Perception Test also uses a single video that is 23 seconds long on average.
>
> Because these datasets rely on a single short video, they do not require cross-video alignment or knowledge fusion across different viewpoints. In contrast, MA-EgoQA uses six videos captured simultaneously from different agents, which **requires models to build a global understanding of the entire multi-agent system while also tracking each agent's perspective.** In addition, MA-EgoQA contains extremely long videos for each query, each video stream is 44 hours long on average and 266 hours in total. This setting is closer to real-world embodied agents that continuously record during operation. **Processing and referencing such long duration videos introduces substantial challenges in constructing an effective memory and retrieving the most relevant information for each query.**
>
> We included this comparison and discussion in `Table 1` and `Section 2.2` (Video Question-Answering Benchmarks) of the revision.
>
> ---
> > **Question 1)** Will the dataset be released?
>
> Sure, we will release the MA-EgoQA benchmark and EgoMAS code for future research in this direction.

---

### Official Review · Reviewer_tkxf · 2025-11-10

**Soundness:** 2
**Presentation:** 2
**Contribution:** 2
**Rating:** 4
**Confidence:** 3

**Summary:**

This paper proposed MA-EgoQA, a benchmark for multi-agent egocentric video QA of embodied agent scenarios. It constructs QA datasets grounded in long-horizon video, and categorizes them in 5 categories. It evaluates multiple LLMs and agents, and proposed a new method, EgoMAS.

**Strengths:**

1. the benchmark fills a recognized gap in existing datasets
2. the dataset construction pipeline and quality control is thorough and clearly described
3. the evaluation is comprehensive, containing most of the open-source LLMs, close-source LLMs, and other agents

**Weaknesses:**

1. the dataset is generated based on EgoLife, it would be better if the author could include more other scenarios rather than only EgoLife
2. the proposed EgoMAS is simple, and does not exhibit a significant improvement, the presence of this method is not so meaningful
3. the windowing strategies and retrieval granularities may not be optimized equivalently across these methods
4. the ablation study should verify the contribution of submodules 4W1H and BM25
5. the efficiency of shared memory, which might be the most important part when this method is applied in real-world, is not discussed in this paper

**Questions:**

1. Why ToM is the hardest task, did you conduct any thorough analysis on the possible reasons?

---

> ### Author Response · Authors · 2025-11-22
> **Response to Reviewer tkxf (1/3)**
>
> We sincerely thank you for your detailed and valuable comments. We also greatly appreciate your recognition of the novelty of our benchmark, clear presentation of dataset generation and filtering, and complete evaluation with 16 baselines and EgoMAS. We have carefully considered your review and made every effort to address all your comments and concerns thoroughly.
>
> ---
> > **Weakness 1)** The dataset is generated based on EgoLife, it would be better if the author could include more other scenarios rather than only EgoLife.
>
> We appreciate your valuable suggestion to include scenarios beyond EgoLife videos in our benchmark. We agree that evaluating models in diverse environments would further strengthen the assessment of generalization. However, we would like to clarify that EgoLife is currently **the only publicly available video dataset** providing **long-term, egocentric videos captured simultaneously from multiple agents.**
>
> Existing egocentric benchmarks (e.g., Ego4D, EgoSchema, EgoThink) either lack a multi-agent setting or do not offer days-long video streams across individuals. Moreover, generating multi-agent, days-long egocentric videos requires substantial human and time resources as well as careful consideration of privacy. These factors make **such video datasets extremely scarce**, despite their importance for real-world embodied AI applications.
>
> Meanwhile, EgoLife provides 266 hours of egocentric videos from six people that **cover a wide range of daily activities and environments**. According to the EgoLife paper, the dataset includes 14 activity categories such as social, housekeeping, cooking, and party, and it also contains activities outside the house, including shopping and sightseeing. Given this breadth, we believe that EgoLife offers sufficiently diverse scenarios to evaluate the multi-agent video understanding capabilities of the models.
>
> We included this discussion in `Appendix B` (Limitation and Future Work) of the revision.
>
> ---
> > **Weakness 2)** EgoMAS is simple, and does not exhibit a significant improvement, the presence of this method is not so meaningful.
>
> We thank you for raising this concern, and we acknowledge that there is room to further improve the structure and performance of EgoMAS. We would like to clarify that **we intentionally designed EgoMAS with a straightforward structure as a train-free, simple yet effective baseline** to support future research in this area, especially as this is **the first work to introduce a multi-agent egocentric video QA task**.
>
> In addition, our ablation studies demonstrate that the **design choices in EgoMAS are not trivial and lead to meaningful insights** for effective multi-agent egocentric video QA. As shown in `Table A.1`, our event focused 4W1H memory structure outperforms other shared memory construction strategies, including a naive summary corpus and a graph structure. In `Table A.2`, we also show that a light BM25 retriever, which relies on keyword matching, performs well within our framework without requiring heavy sentence embedding models.
>
> For the comparison with baselines, we would like to highlight **the strength of EgoMAS in the balance between accuracy and efficiency**. As shown in `Table A.3`, EgoMAS (Text) achieves 35.55% accuracy with around 1.3 sec/query, while Gemini-2.5-Flash obtains 36.93% accuracy with around 50.0 sec/query latency. Compared to other baselines, EgoMAS demonstrates the most favorable position in the trade-off between latency and accuracy, as illustrated in `Figure 6` of the revision. Furthermore, **EgoMAS employs a small open source model** (Qwen2.5-VL 7B) for retrieval reasoning and response generation, which **improves its practicality as a future baseline**.
>
> We included this discussion in `Section 6` of the revision to highlight the effectiveness and contributions of two modules in EgoMAS.
>
> **Table A.1.** Ablation study on the structure of shared memory.
> |Shared Memory Structure|Acc.|
> |-|-|
> |Summary|30.67|
> |Triplet|30.44|
> |Chunk|25.96|
> |Graph|31.99|
> |Ours (4W1H)| **35.55** |
>
> **Table A.2.** Ablation study on the memory retriever.
> |Memory Retriever| Acc.|
> |-|-|
> |DPR| 28.66 |
> |Qwen3 Embed-0.6B| 33.03 |
> |NV-Embed-v2 7B| **37.91** |
> |Ours (BM25)| _35.55_ |
>
> **Table A.3.** Analysis on efficiency of models.
> |Models|Avg. Latency(s)| Avg. # Input Tokens | Acc.|
> |-|-|-|-|
> | **Text-only**|
> |Gemini-2.5-Flash|49.97|1M|36.93|
> |Qwen2.5-7B-Instruct-1M|3.56|500k|26.08|
> |GPT-5|33.58|272k|34.81|
> |**Video-only**|
> |Qwen2.5-VL-7B|69.80|128k|25.22|
> |VideoChat-Flash|19.55|32k|23.06|
> |**Retrieval-based**|
> |BM25|12.17|11.55k|33.49|
> |EgoRAG|2.59|0.415k|19.57|
> |EgoMAS (Text)|1.30|3.01k|35.55|

---

> ### Author Response · Authors · 2025-11-22
> **Response to Reviewer tkxf (2/3)**
>
> > **Weakness 3)** Windowing strategies and retrieval granularities may not be equally optimized across methods.
>
> We appreciate you raising this point and giving us the opportunity to clarify the evaluation setting used for the models. **The windowing strategy is applied only in EgoMAS**, since the baseline models do not leverage a shared memory structure. For text-only and video-only baselines, we simply concatenate all information from all agents and use it as the input context. For the retrieval-based baselines, BM25, DPR, and InternVideo2 operate on a single merged corpus that contains all information from all agents, and the retriever selects the relevant memory from this unified source. In EgoRAG and Ego-R1 baselines, we follow their original model design, which uses only a single agent memory.
>
> Regarding retrieval granularities, **we also follow the original granularity configurations** of EgoRAG and Ego-R1, which use levels of **30 seconds, 10 minutes, 1 hour, and 1 day**. For other retrieval-based baselines, we use a **30-second clip** as the retrieval unit. In the case of EgoMAS, the shared memory is retrieved at the granularity of **10-minute captions**, and the agent-wise dynamic retrieval is performed at the granularity of **30-second segments**.
>
> ---
> > **Weakness 4)** The ablation study should verify the contribution of submodules 4W1H and BM25.
>
> Thank you for highlighting this important analysis that was missing in the paper. We validate the contribution of each submodule through ablation studies and present above in `Table A.1` and `Table A.2`.
>
> To assess the effectiveness of 4W1H memory, we evaluate multiple memory construction strategies introduced in prior works [1,2]. As shown in `Table A.1`, **the event-based 4W1H memory structure clearly outperforms alternative methods**, demonstrating its strong ability to abstract and fuse the events across multiple agents.
>
> In `Table A.2`, we evaluate the choice of memory retriever. While NV-Embed-v2 achieves the highest accuracy, it has 7B parameters and thereby introduces significant computational and temporal overhead. In contrast, **BM25 operates through lightweight keyword matching and achieves competitive performance while maintaining much lower cost**, which makes it a more practical choice for our framework.
>
> We included these results in `Table 4` of the revision to clearly show the contribution of 4W1H shared memory and BM25 retrieval.
>
> [1] Zeng et al., On the structural memory of llm agents, preprint 2024
>
> [2] Gutiérrez et al., From rag to memory: Non-parametric continual learning for large language models, ICML 2025
>
> ---
> > W5. The efficiency of shared memory, which might be the most important part in real-world application, is not discussed in the paper.
>
> We fully agree that efficiency is crucial for real-world applications, and we greatly appreciate you raising this point. In `Table A.3`, we report the average latency and input token length measured on randomly selected 100 samples from MA-EgoQA across several baseline models and EgoMAS. As we mentioned earlier, non-retrieval-based models incur significantly high overhead, while retrieval-based models, including EgoMAS, show notably lower latency and smaller input sizes. In particular, EgoMAS achieves the highest accuracy among retrieval-based models while requiring **only 1.3 seconds per query**. These results highlight that **EgoMAS is a practical approach for building a multi-agent egocentric video QA system in real world settings.**
>
> We included these results and analysis in `Figure 5` and `Section 6` of the revision to present the remarkable efficiency of EgoMAS.

---

> ### Author Response · Authors · 2025-11-22
> **Response to Reviewer tkxf (3/3)**
>
> > Q1. Why ToM is the hardest task, did you conduct any thorough analysis on the possible reasons?
>
> **ToM questions require models to infer beliefs, intentions, or perspectives that are not directly observable in videos or transcripts.** While questions in other categories are typically grounded in explicit actions or conversations, ToM questions demand reasoning about the internal mental state, which must be inferred from subtle cues across multiple agents. This inherently requires strong multi-agent knowledge fusion and a better understanding in a system-wise context.
>
> Consistent with this, the ToM category shows the lowest average accuracy across all evaluated models, indicating that existing approaches struggle with this type of reasoning. To further illustrate the challenge, we present an additional example below.
> ```
> [ ToM Example 1 ]
> Q. What did Katrina wrongly assume while looking at Nicous and Violet filming?
> A. That they were wrapping up their session immediately.
> ```
> In this example, the model must first compare Katrina’s memory with the actual filming context of Nicous and Violet, identify the discrepancy between their perspectives, and then infer the mistaken belief. **This demonstrates that ToM questions require deeper inference and more effective integration of multi-agent information than other categories.**
> ```
> [ ToM Example 2 ]
> Q. Why did Shure think Choiszt might have misinterpreted the coffee instructions?
> A. Choiszt confused the steps and thought milk was required.
>
> [ ToM Example 3 ]
> Q. Why didn't Katrina know where to put the big kitchen tools?
> A. Shure had placed things randomly, making it hard to locate spaces.
> ```
> These examples further illustrate that **the model must go beyond recognizing observable actions or dialogue and instead infer the underlying mental states of the agents.** In Example 2, the conversation between Shure and Choiszt about the coffee instructions does not explicitly reveal Choiszt’s misunderstanding, and the model must infer it from subtle cues in their interaction.
> In Example 3, Katrina’s lack of knowledge about where to place the kitchen tools is not directly stated; rather, the model must connect her confusion to Shure’s earlier action of placing items randomly. These cases highlight that ToM questions require additional layers of reasoning that extend beyond what is directly visible or spoken.

---

### Author Response · Authors · 2025-11-26
**General Response to Reviewers**

We sincerely appreciate the reviewers' time and effort in reviewing our paper. We are encouraged by their acknowledgements, including 1) the **novelty of MA-EgoQA** as the first benchmark introducing multi-agent long egocentric video QA which is highly crucial for real-world application of embodied AI [tkxf, AiKH, 8MQJ, oPM5], 2) the **effectiveness of the simple yet strong EgoMAS framework**, which outperforms most existing models and demonstrate its practicality as a baseline for future research [AiKH, 8MQJ], 3) the **clear presentation** of our category-wise dataset construction and careful quality control [tkxf, AiKH] supported by **human validation efforts** [oPM5], and 4) the **comprehensive evaluation with 16 baselines** and EgoMAS on MA-EgoQA, which highlights the limitations of current single-agent focused models in our task and paves the way for improving architectures and knowledge fusion methods in multi-agent system [tkxf, AiKH].

During the rebuttal, we clarify the following points from the comments:
- **Event-based shared memory and agent-wise dynamic retrieval based on BM25 outperform their corresponding simple baselines. [tkxf, 8MQJ]** We conduct additional experiments with multiple shared memory structures and different retrieval strategies, and verify the contribution of each module in EgoMAS.
- **The experiment settings for all baselines follow their original configurations, which are similar to those used in EgoMAS. [tkxf, oPM5]** Two retrieval-based models use 30-second, 10-minute, 1-hour, and 1-day granularity, while the other retrieval models use 30-second segments for retrieval. Similarly, EgoMAS uses 10-minute captions for constructing shared memory and 30-second captions for agent-wise retrieval.
- **Theory-of-Mind questions are difficult since they require additional reasoning beyond the provided visual and conversational contexts. [tkxf, 8MQJ]** ToM questions reflect both semantic knowledge from conversations and visuo-cognitive information, as they are generated using transcripts and visually dense captions. These aspects are discussed with example QA samples in the rebuttal.
- **The small difference between EgoMAS (Text+Video) and EgoMAS (Text) arises from overlapping information between textual and visual modalities. [AiKH, oPM5]** This observation suggests a future research direction on leveraging visual information alongside abstracted text memory, whereas EgoMAS primarily focuses on memorization and retrieval strategies as a strong future baseline.
- **The efficiency analysis highlights the remarkable practicality of EgoMAS. [tkxf]** We compare the average latency and input token lengths of baselines and EgoMAS, and EgoMAS shows the most favorable latency-accuracy trade-off, requiring only 1.3 seconds per query while Gemini-2.5-Flash takes around 50 seconds per query.
- **MA-EgoQA is a genuinely multi-agent benchmark, as shown by the experiments varying the number of available and required agents. [8MQJ]** We observe that the performance improves as the number of available agents increases, while accuracy degrades when a query requires memories from a large set of agents, underscoring the benchmark's dependence on multi-agent memory retrieval.

We revised the paper based on the reviewers' feedback and enhanced its clarity as follows:
- We included ablation studies on two modules of EgoMAS in `Table 4` and their discussion in `Section 6` to clarify the contribution of each design.
- We added the performance trends with respect to the number of available agents and the number of required agents in `Figure 4` and `Figure 5`, and analysed these results in `Section 6` to clarify our benchmark's dependence on multi-agent retrieval.
- We illustrated the latency-accuracy trade-off of baselines and EgoMAS in `Figure 6` and discussed the significant efficiency of our model in `Section 6`.
- We included additional related works and compared them with MA-EgoQA in `Table 1` and `Section 2` to clarify the differences and the novel challenges addressed by MA-EgoQA.
- We provided more qualitative QA examples of MA-EgoQA in `Appendix C` to facilitate a clear understanding of the benchmark.
- We added the limitations and future work of this research in `Appendix B` to clarify the constraints of our research and to propose meaningful directions for future studies.

We greatly appreciate the constructive comments from the reviewers. We kindly encourage the reviewers to consider our rebuttal, including the clarifications and additional experiments we have presented. We are happy to address any remaining concerns and welcome further discussion to strengthen our work.

---

### Meta-Review · Area_Chair_rHd3 · 2026-01-07

**Summary:**

The paper presents a method for ego-centric video understanding for multiple embodied agents. The submission receives mixed ratings from four reviewers. Most of them provided negative ratings and raised several critical concerns regarding the technical contribution and the experimental verification. These concerns include the unclear motivation and effectiveness of the key modules in the proposed framework, the insufficient performance improvements over the baseline model, the lack of evaluation on some important benchmarks, and the missing discussion on the efficiency for real-world applications. The rebuttal can only partially address these critical issues. Based on the overall scores and the comments, AC finally decided to recommend a rejection of this submission this time.

**Reviewer Concerns:**

The major concerns from the reviewer are summarized as follows: (i) the unclear motivation and effectiveness of the key modules in the proposed framework, (ii) the insufficient performance improvements over the baseline model, (iii) the lack of evaluation on some important benchmarks, and (iv) the missing discussion on the efficiency for real-world applications. The rebuttal did not fully convince the reviewers, and AC believes that significant efforts are necessary to make the paper ready.

**Reviewer Scores:**

The initial reviewer score distribution is 4, 8, 4, 2. The reviewers did not show intentions to increase their scores. The decision is thus straightforward for this submission.

---

### Decision · Program_Chairs · 2026-01-26

Reject